# Adaptive Visual Scene Understanding: Incremental Scene Graph Generation

**Naitik Khandelwal**[1,2]**, Xiao Liu**[1,2] **Mengmi Zhang**[1,2]

[1] College of Computing and Data Science, Nanyang Technological University (NTU), Singapore

[2] Deep NeuroCognition Lab, Agency for Science, Technology and Research (A*STAR), Singapore

Address correspondence to mengmi.zhang@ntu.edu.sg

## Abstract

Scene graph generation (SGG) analyzes images to extract meaningful information about objects and their relationships. In the dynamic visual world, it is crucial for AI systems to continuously detect new objects and establish their relationships with existing ones. Recently, numerous studies have focused on continual learning within the domains of object detection and image recognition. However, a limited amount of research focuses on a more challenging continual learning problem in SGG. This increased difficulty arises from the intricate interactions and dynamic relationships among objects, and their associated contexts. Thus, in continual learning, SGG models are often required to expand, modify, retain, and reason scene graphs within the process of adaptive visual scene understanding. To systematically explore Continual Scene Graph Generation (CSEGG), we present a comprehensive benchmark comprising three learning regimes: relationship incremental, scene incremental, and relationship generalization. Moreover, we introduce a "Replays via Analysis by Synthesis" method named RAS. This approach leverages the scene graphs, decomposes and re-composes them to represent different scenes, and replays the synthesized scenes based on these compositional scene graphs. The replayed synthesized scenes act as a means to practice and refine proficiency in SGG in known and unknown environments. Our experimental results not only highlight the challenges of directly combining existing continual learning methods with SGG backbones but also demonstrate the effectiveness of our proposed approach, enhancing CSEGG efficiency while simultaneously preserving privacy and memory usage. All data and source code are publicly available here.

## 1 Introduction

Scene graph generation (SGG) aims to extract object entities and their relationships in a scene. The resulting scene graph, carrying semantic scene structures, can be used for a variety of downstream tasks such as object detection[64], image captioning [20, 1] , and visual question answering [17]. Despite the notable advancements in SGG, current works have largely overlooked the critical aspect of continual learning. In the dynamic visual world, new objects and relationships are introduced incrementally, posing challenges for SGG models to account for new changes without forgetting previously acquired knowledge. This problem of Continual ScenE Graph Generation (CSEGG) holds great potential for various applications, such as real-time robotic navigation in dynamic environments and adaptive augmented reality experiences.

The field of continual learning has witnessed significant growth in recent years, with a major focus on tasks such as image classification [43], object detection [67], and visual question answering [28]. However, these endeavors have largely neglected the distinctive complexities associated with CSEGG. Here, we highlight several unique challenges of CSEGG: (1) In contrast to object detection,

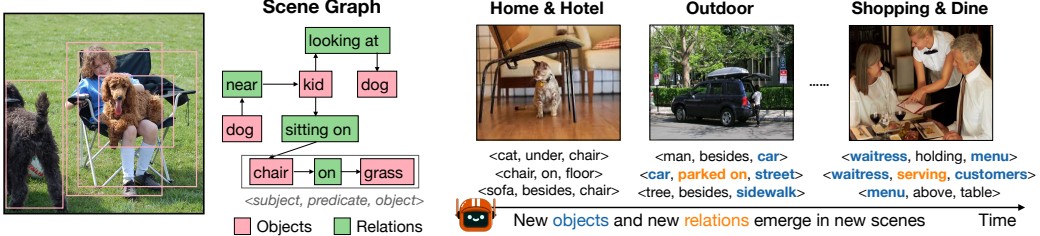

Figure 1: (a) **A scene graph is a graph structure,** where objects are represented as nodes (red boxes), and the relationships between objects are represented as edges connecting the corresponding nodes (green boxes). Each node in the graph contains information such as the object's class label, and spatial location. The edges in the graph indicate the relationships between objects, often described by predicates. A scene graph can be parsed into a set of triplets, consisting of three components: a subject, a relationship predicate, and an object that serves as the target or object of the relationship. The graph allows for a compact and structured representation of the objects and their relationships within a visual scene. (b) **An example CSEGG application** is presented, where a robot continuously encounters new objects (blue) and new relationships (yellow) over time across new scenes.

SGG involves understanding and capturing the relationships between objects, which can be intricate and diverse. Consequently, in CSEGG, conveying the spatial and semantic relationships between objects demands adaptive reasoning from the dynamic scene. (2) SGG introduces a higher level of combinatorial complexity than object detection and image classification because each detected object pair may have multiple potential spatial and functional relationships. Thus, as new objects are introduced to the scenes, the complexity of relationships among all the objects increases significantly in a non-linear fashion. (3) The long-tailed distribution in both objects and relationships in SGG can be attributed to the inherent characteristics of real-world scenes, where certain objects are more prevalent than others. Consequently, CSEGG requires the computational models to adapt continually to the evolving long-tailed distributions over different scenes. Due to a scarcity of research specifically addressing these challenges of CSEGG, there is a pressing need for specialized investigations and methodologies to enable computational models with the ability of CSEGG.

In this study, we re-organize existing SGG datasets [25, 27] to establish a novel and comprehensive CSEGG benchmark with 3 learning protocols as shown in **Fig. 2**. **(S1).** Relationship-incremental setting: an SGG agent learns to recognize new relationships among familiar objects within the same scene. **(S2).** Scene-incremental setting: an SGG agent is deployed in new scenes where it has to jointly learn to detect new objects and classify new relationships. **(S3).** Relationship generalization setting: an SGG agent generalizes to recognize known relationships among unknown objects, as the agent learns to recognize new objects.

We curate a set of competitive CSEGG baselines by directly combining three major categories of continual learning methods with two SGG backbones and benchmark them in our CSEGG dataset. Their inferior performances show the difficulties of our benchmark tasks, which require the ability to expand, modify, retain, and reason scene graphs within the process of adaptive visual scene understanding. Specifically, the weight-regularization methods fail to estimate the importance of learnable parameters given the complicated model design in SGG backbones. Although image-replay methods retain knowledge from prior tasks through replays, the extensive combinatorial complexity of relationships among objects surpasses the complexity accommodated by a restricted set of replay images with efficient storage. Additionally, none of these baseline methods consider the shifts inherent in long-tailed distributions in dynamic scenes.

To address the CSEGG challenges, we present a method called "Replays via Analysis by Synthesis", abbreviated as RAS. RAS employs scene graphs from previous tasks, breaks them down and re-composes them to generate diverse scene structures. These compositional scene graphs are then used for synthesizing scene images for replays. Due to its nature of symbolic replays, RAS does not require the storage of original images, which often carry excessive and redundant details. This also ensures data privacy preservation and data efficiency. Furthermore, by synthesizing scenes using composable scene graphs, RAS maintains the semantic context and structure of previous scenes and also enhances the diversity of scene generation. To prevent biased predictions stemming from long-tailed distributions, we moderate the distribution of replayed scene graphs by balancing tail and head classes. This ensures a uniform sampling of relationships and objects during replays. Extensive

| S. | #Tasks | #Objs | #Rels | Eval. metrics | SGG Backbone | CL base. | Kn. | Unk. |
|---|---|---|---|---|---|---|---|---|
| **S1** | 5 | 150 (*All*) | 10 *per task* | F, R, mF, mR, FWT, BWT, Gen R$_{bbox}$, Gen R | Transformer based (SGTR) | Joint, Naive, Replay M%, EWC, PackNet, RAS_GT | Objs (bbox, labels) | Rels |
| **S2** | 2 | *Task 1*: 100 *Task 2*: 25 | *Task 1*: 40 *Task 2*: 5 | | CNN based (IMP) | | None | Rels and Objs (bbox, labels) |
| **S3** | 4 | 30 *per task* | 35 *per task* | | | | Rels | Objs (bbox, labels) |

Table 1: **Overview of three CSEGG learning scenarios.** This table summarizes the three learning scenarios (Column 1) in CSEGG, including the number of tasks, the number of object (#Objs) and relationship (#Rels) classes, the evaluation metrics, the SGG-Backbones used, and the continual learning (CL) baselines. The Kn. and Unk. columns provide information regarding what is known to the CSEGG models during training in that scenario and what is being incrementally learned by the models. Unknown information is being incrementally learned by the models. See **Sec. 3** for details.

experiments underscore the effectiveness of our approach. Network analysis reveals our crucial design choices that can be beneficial for the future development of CSEGG models.

## 2 Related Works

**Scene Graph Generation Datasets.** Visual Phrase [59] stands as one of the earliest datasets in the field of visual phrase recognition and detection. Over time, various large-scale datasets have emerged to tackle the challenges of Scene Graph Generation (SGG) on static images [23, 42, 25, 27, 37, 78, 74, 72, 80, 12, 35, 81]. Subsequent works further extend the SGG to dynamic videos [22, 50, 56]. Despite the significant contributions of these datasets to SGG, none focuses on continual learning in SGG. As the preliminary efforts towards CSEGG, we start with fundamental and straightforward settings of SGG on static images. Among all the SGG datasets on static images, the Visual Genome dataset [25] has played a pioneering role by providing rich annotations of objects, attributes, and relationships in images. Thus, we re-structure the Visual Genome dataset [25] and establish a novel and comprehensive CSEGG benchmark, where AI models are deployed to dynamic scenes where new objects and new relationships are introduced.

**Scene Graph Generation (SGG) Models.** SGG models are categorized into two main approaches: top-down and bottom-up. Top-down approaches[38, 77] typically rely on object detection as a precursor to relationship prediction. They involve detecting objects and then explicitly modeling their relationships using techniques such as rule-based reasoning[42] or graph convolutional networks [73]. On the other hand, bottom-up approaches focus on jointly predicting objects and their relationships in an end-to-end manner [34, 35, 72]. These methods often employ graph neural networks [33, 82] or message-passing algorithms [72] to capture the contextual information and dependencies between objects. Furthermore, recent works have explored the integration of language priors [48, 42, 69] and attention mechanisms in transformers [3] to enhance the accuracy and interpretability of scene graph generation. However, none of these works evaluate SGG models in the context of continual learning. In our work, we directly combine continual learning methods with SGG backbones and benchmark these competitive baselines in CSEGG. Our results reveal the limitations of these methods and highlight the challenges of our CSEGG learning protocols.

**Continual Learning Methods.** Existing continual learning works can be categorized into several approaches. (1) Regularization-based methods [24, 9, 79, 2, 4] aim to mitigate catastrophic forgetting by employing regularization techniques in the parameter space. (2) Dynamic architecture-based approaches[66, 76, 21, 47] adapt the model's architecture dynamically to accommodate new tasks without interfering with the existing ones. (3) Replay-based methods [57, 10, 55, 65, 52, 7] utilize a memory buffer to store and replay past data during training, enabling the model to revisit and learn from previously seen examples, thereby reducing forgetting. The special variants of these methods include generative replay methods, such as [61, 71, 75, 51], where synthetic data is generated and replayed. Although these generative replay methods, as well as other continual learning methods, have been extensively studied in image classification [8, 70, 43] and object detection[68, 60, 45], few works focus on the challenges in CSEGG, such as adaptive reasoning from the dynamic scenes, the evolving long-tailed distribution across scenes, and the combinatorial complexity involving objects and their multiple relationships. In this work, we introduce a continual learning method, abbreviated as RAS (Replays via Analysis by Synthesis). To address the distinct challenges of CSEGG, RAS

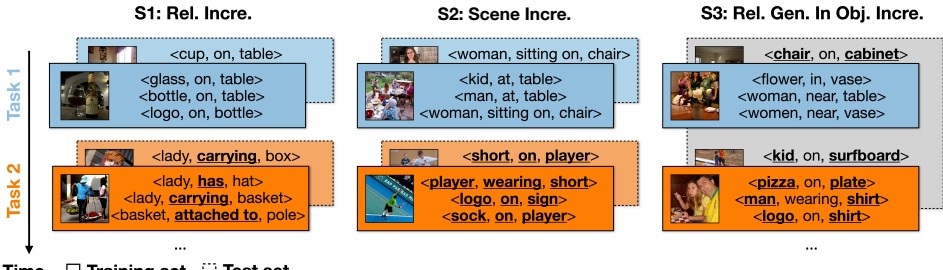

Figure 2: **Three learning scenarios** are introduced. From left to right, they are S1. relationship (Rel.) incremental learning (Incre.); S2. scene incremental learning; and S3. relationship generalization (Rel. Gen.) in Object Incre.. In S1 and S2, example triplet labels in the training (solid line) and test sets (dotted line) from each task are presented. The training and test sets from the same task are color-coded. Blue color indicates task 1 and orange color indicates task 2. The new objects or relationships in each task are bold and underlined. In S3, one single test set (dotted gray box) is used for benchmarking the relationship generalization of object incre. learning models across all the tasks.

involves creating in-context synthetic scene images based on re-composable scene graphs from previous tasks to reinforce continual learning. The components in RAS facilitate memory-efficient training and preserve privacy while maintaining the scene diversity and scene context for SGG in dynamic environments. With the rise of pretrained vision-language models (VLMs) [49, 31, 83, 40], various SGG methods [11, 30] have been proposed to tackle open-vocabulary and zero-shot SGG challenges. However, these settings differ fundamentally from CSEGG, where we aim to simulate scenarios where the model encounters novel predicates or objects, unseen by any models, including LLMs or multi-modal models. Using pre-learned information from LLMs or frozen encoders conflicts with the continual learning setting we address in CSEGG.

## 3 Continual ScenE Graph Generation Benchmark

In CSEGG, to cater to the three continual learning scenarios below, we re-organize the Visual Genome [25] dataset and follow its standard image splits for training, validation, and test sets specified in [72]. In each learning scenario, we consider a sequence of $T$ tasks consisting of images and corresponding scene graphs with new objects, or new relationships, or both. Let $D_t = \{(I_i, G_i)\}_{i=1}^{N_t}$ represent the dataset at task $t$, where $I_i$ denotes the $i$-th image and $G_i$ represents the associated scene graph. The scene graph $G_i$ comprises a set of object nodes $O_i$ and their corresponding relationships $R_i$. Each object node $o_j$ is defined by its class label $c_j$ and its bounding box locations and sizes $b_j$. Each relationship $r_k$ is represented by a triplet $(o_s, p_k, o_o)$, where $o_s$ and $o_o$ denote the subject and object nodes, and $p_k$ represents the relationship predicate.

### 3.1 Learning Scenarios

**Scenario 1 (S1): Relationship Incremental Learning.** To uncover contextual information and go beyond studies of object detection and recognition, we introduce this scenario consisting of 5 tasks where 10 new relationship classes are incrementally added in every task (**Fig. 2, left**; **Fig. S1**; **Tab. 1**). All object classes and their locations are made known to all CSEGG models over all the tasks. This scenario resembles a human learning scenario where a parent gradually teaches a baby to recognize new relationships among all objects in the same room, focusing on one new relationship at a time during continual learning. This scenario also has implications in medical imaging where identical cell types may form new relationships with nearby cells depending on the context (**Sec. A.1.1**).

**Scenario 2 (S2): Scene Incremental Learning.** To simulate the real-world cases when there are demands for detecting new objects and new relationships from old to new scenes, we introduce this scenario where new objects and new relationships are incrementally introduced over tasks (**Fig. 2, middle**; **Fig. S1**; **Tab. 1**). There are 2 tasks in total with the first task containing 100 object classes and 40 relationship classes with 25 more object classes and 5 more relationship classes in the second task. This aligns with the real-world use cases where common objects and relationships are learned in the first scene, and incremental learning in the second scene only happens on less frequent relationships and objects. See **Sec. A.1.2** for details.

**Scenario 3 (S3): Relationship Generalization.** Humans have no problem at all recognizing the relationships of unknown objects with other nearby objects. This scenario is designed to investigate the relationship generalization ability of CSEGG models. This capability is essential for real-world implications, such as in robotic navigation where it often encounters unknown objects and requires classifying their relationships. In total, there are four tasks, each introducing an incremental addition of 30 new object classes. All relationship classes are made known to all CSEGG models over all the tasks (**Fig. 2, right**; **Fig. S1**; **Tab. 1**). Different from scenarios **S1** and **S2**, a standalone generalization test set is curated, where the objects are unknown but the relationship classes among these unknown objects are common to the training set of every task. The CSEGG models trained after every task are tested on this standalone generalization test set to predict relationships among the unknown objects. See **Sec. A.1.3** for details.

**Data sampling and distributions.** To allocate data for every task of each scenario, we perform the following sampling strategies. In **S1** and **S3** above, either object or relationship classes are randomly sampled from the Visual Genome dataset and incrementally added to every task. Due to the inherent characteristics of real-world scenes, the long-tailed class distribution is present in **S1** and **S3**. However, in **S2**, only tail classes are sampled and added in subsequent tasks. The number of tasks in each scenario is experimentally determined to optimize the training data configuration, ensuring sufficient training samples in each task while maximizing the number of tasks. For detailed statistics, see **Fig. S2** and **Sec. A.2**.

## 3.2 Competitive CSEGG Baselines

Due to the scarcity of CSEGG works, we contribute a diverse set of competitive CSEGG baselines and implement them on our own. Each CSEGG baseline requires three components: a backbone model for scene graph generation (SGG), a continual learning (CL) method to prevent the SGG model from forgetting, and an optional data sampling technique to deal with imbalanced data at every task for training SGG models. Next, we introduce the 2 SGG backbones, the 5 continual learning methods, and the 5 optional data sampling techniques. See **Sec. A.3** for implementation and training details of CSEGG baselines.

**SGG Backbones.** We use the two state-of-the-art backbones: (1) one-stage Scene graph Generation TRansformer (SGTR) [32] and (2) the traditional Two-stage SGG model (TCNN) [72]. Briefly, SGTR (**Fig. S3 left**) uses a transformer-based architecture for image feature extraction and fusion. During training, [32] formulates SGG as a bipartite graph construction and matching problem. In contrast, TCNN detects objects with Faster-RCNN[18] backbone and infers their relationships separately via Iterative message passing [72]. We use implementations from [32] and [68] with default hyperparameters.

**Baselines.** We include the following continual learning methods (**Fig. S3 right**): (1) Naive (lower bound) is trained on each task in sequence without any measures to prevent catastrophic forgetting. (2) EWC[24] is a weight-regularization method, where the weights of the network are regularized in the parameter space, based on their "importance" to the previous tasks. (3) PackNet[44] is a parameter-isolation method, iteratively pruning the network parameters after every task, so that it can sequentially pack multiple tasks within one network. (4) Replay@M[57] includes a memory buffer with the capacity of storing $M$ percentages of images in the entire dataset as well as their corresponding ground truth object and predicate notations depending on the task at each learning scenario. We vary $M = 10\%$, $20\%$, and $100\%$. (5) Joint Training is an upper bound where the SGG model is trained on the entire CSEGG dataset. (6) RAS_GT is a baseline in which we use the ground truth scene graph labels from each task to create replay buffers using an image generation model explained in detail in **Sec. 4**. See **Fig. S4** for schematics of CSEGG baselines. We provide mathematical formulations of these baselines in **Sec. A.4**.

**Sampling Methods to Handle Long-Tailed Distribution.** We adopt the five data sampling techniques to alleviate the problem of imbalanced data distribution during training. (1) LVIS[19] is an image-level over-sampling strategy for the tailed classes. (2) Bi-level sampling (BLS) [33] balances the trade-off between image-level oversampling for the tailed classes and instance-level under-sampling for the head classes. (3) Equalized Focal Loss (EFL) [29] is an effective loss function, re-balancing the loss contribution of head and tail classes according to their imbalanced distribution. EFL is enabled all the time for all the CSEGG baselines. In addition to applying data sampling techniques to the training sets, we can also apply LVIS and BLS techniques to the data stored in the

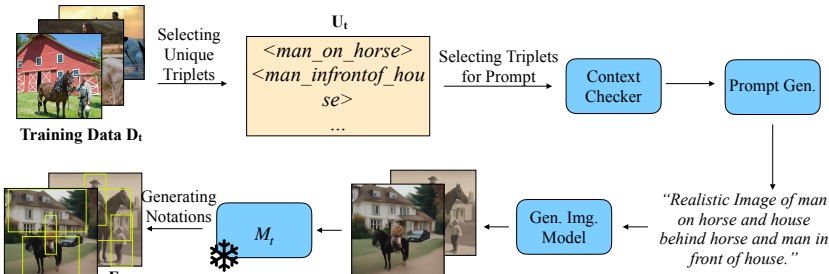

Figure 3: **Schematic of our proposed Replays via Analysis by Synthesis (RAS) method.** At task $t+1$, our RAS stores all the triplet labels $U_t$, such as <man, on, horse>, from the previous tasks. It then re-composes these triplet labels to create in-context prompts, utilizing them as inputs to generative image models to synthesize images for replays. For predicting scene graphs on these synthesized images, we employ the frozen model $M_t$ from the preceding task $t$, marked with "snowflakes". Subsequently, these predicted scene graph notations, along with their corresponding synthesized images, contribute to "pseudo" replays, preventing the current model $M_{t+1}$ from experiencing forgetting. See **Sec. 4** for more details.

replay buffer. We name these data sampling techniques applied during both training and replays as (4) LVIS@Replay and (5) BLS@Replay.

### 3.3 Evaluation Metrics

Same as existing SGG works [72, 32], we adopt the evaluation metric recall@K (**R@K**) on the top $K$ predicted triplets in the scene graphs $G$. As CSEGG is long-tailed, we further report the results in mean recall (**mR@K**) over the head, body, and tail classes. Forgetfullness (F), Average (Avg.) performance, Forward Transfer (FWT) [39] and Backward Transfer (BWT) [41] are standard evaluation metrics used for continual learning in image recognition and object detection tasks. In Scenario 1 and 2, we adapt these metrics to recalls R@K and introduce **F@K**, **Avg.R@K**, **FWT@K**, and **BWT@K** respectively for CSEGG settings. Similarly, we also adapt these metrics to mR@K. We explored CSEGG with K=20, 50, and 100. Since our results are consistent among Ks, we omit "@K" and analyze all the results based on K=20 in the entire text.

In scenario S3, we evaluate all CSEGG methods in the standalone generalization test set, shared over all the tasks. To benchmark generalization abilities in unknown object localization and relationship classification among these unknown objects, we introduce two evaluation metrics: **Gen $R_{bbox}$@K** and **Gen R@K**. As the CSEGG models have never been taught to classify unknown objects, we discard the class labels of the bounding boxes and only evaluate the predicted bounding box locations with **Gen $R_{bbox}$@K**. To evaluate whether the predicted bounding box location is correct, we apply a hard threshold of Intersection over Union (**IoU**) between the predicted bounding box locations and the ground truth. Any predicted bounding boxes with their IoU values above the hard threshold are deemed to be correct. We vary IoU thresholds from 0.3, 0.5, to 0.7.

To assess whether the CSEGG model generalizes to detect known relationships over unknown objects, we evaluate the recall **Gen R@K** of the predicted relationships $r_k$ only on *correctly predicted* bounding boxes. See **Sec. A.5** for details. All results are averaged over 3 runs.

## 4 Replays via Analysis by Synthesis (RAS)

To address the complexities in CSEGG, we introduce our "Replays via Analysis by Synthesis" method, dubbed RAS. Our RAS belongs to the group of generative replay methods for continual learning. We create an exemplar set $E_t$ for replays. At task $t$, we jointly train the scene graph generation model $M_t$ on $E_t$ and the current training dataset $D_t$. However, different from the existing generative replay methods [61, 15, 16], our RAS leverages symbolic replays with state-of-the-art diffusion models. Moreover, rather than generating any random images for replays, our RAS is capable of generating in-context images following semantic rules, such as object co-occurrences. The schematic of our RAS is presented in **Fig. 3**. Next, we focus on how RAS creates $E_t$ containing the generated images and their SGG annotations on these images for replays.

**Image Generation.** At the current task $t+1$, our RAS requires storing the frozen old model snapshot $M_t$ at the end of the previous task $t$ and all the triplet labels $U_t$, which are parsed from all the scene graphs aggregated from all the previous tasks. These triplets contain object labels, subject labels, and relationships among them. For example, <man, on, horse> and <man, in front of, horse> are two unique triplet labels. Unlike the traditional replay methods in continual learning literature, our method refrains from storing original images $I_i$ or scene graphs $G_i$ in the training sets, thereby eliminating storage issues and privacy concerns.

To generate images $I'_j$ for replays, our RAS feeds text prompts, which are formed by a set of chosen triplet labels and describe the diverse in-context scenes, into the state-of-the-art Stable Diffusion model [58]. As previous works suggests context plays important roles in visual perceptions [5]. To generate text prompts describing context-congruent scenes, we employ a context checker. First, the context checker uses the pre-trained large language model BERT [14] to extract embeddings for each triplet label in $U_t$. As BERT has been pre-trained on a large corpus of text data, it learns to capture context-relevant representations of words. Next, hierarchical clustering is performed on these embeddings using the agglomerative clustering algorithm [46]. This ensures that each cluster contains only embeddings that are semantically close. The threshold for the agglomerative clustering algorithm is set to 0.6. As real-world images often contain complex scenes involving multiple triplets, we select any cluster with more than 3 triplet labels to create a text prompt for image generation.

In practical applications, conducting agglomerative clustering on all triplet labels in $U_t$ is computationally demanding, as it requires computing pairwise embedding similarities among all the triplet labels. To address this, RAS opts for a more efficient approach during replays by selecting a subset of triplet labels and clustering their embeddings. Recognizing that real-world scenes often exhibit a long-tailed distribution with certain objects or relationships being more prevalent, we introduce the Long-Tailed Distribution (LTD) module for balancing this distribution in $U_t$. Unlike image-level sampling methods like BLS and LVIS [33, 19] discussed in **Sec. 3.2**, our LTD module in RAS operates at the triplet level. For each triplet label, its dropout rate is determined proportionally to its frequency in $U_t$. Specifically, we define the drop-out rate $d_k$ for the $k$-th triplet as: $d_k = f_k/(\sum_{i=1}^{i=N} f_i) * \alpha$, where $N$ is the total number of triplets in $U_t$, $\alpha = 0.7$ is a scaling factor, and $f_i$ is the frequency of the $i$-th triplet in $U_t$. This sampling formula enables RAS to select triplets from tail classes more frequently compared to those from head classes.

To generate a text prompt from the chosen triplet labels, we employ a straightforward English language construct using the conjunction "and". This involves combining all the selected triplet labels into a sentence by starting with "Realistic Image of". For instance, if the triplet labels are <man, on, horse>, <house, behind, horse>, and <man, in front, house>, the generated prompt becomes "Realistic Image of man on horse and house behind horse and man in front of house." To increase exemplar diversity for replays, we use the Stable Diffusion model [58] to generate $\gamma$ number of images for the same text prompt. In practice, we set $\gamma = 10$ over all the learning scenarios.

We provide the visualization examples of some synthesized images along with the corresponding text prompts in **Fig. S5**. From these examples, we found that the composed text prompts and the synthesized images are often of high quality and contextual coherence.

**Scene Graph Prediction on Synthesized Images.** During replays, to train the model $M_{t+1}$ on $I'_j$, we also need to predict their corresponding scene graph notations $G'_j$ on $I'_j$. As the frozen model snapshot $M_t$ at the end of task $t$ carries prior knowledge for SGG from the previous tasks, we use it to predict notations $G'_j$ on $I'_j$. These $G'_j$ comprises object nodes $O'_j$ with their respective classes $c'_j$, along with object locations $b'_j$. Additionally, it includes corresponding relationship nodes $R'_j$ formed by triplets <$o'_s, p'_k, o'_j$> representing subject, predicate, and object nodes, respectively. These generated notations $G'_j$, along with $I'_j$, serve to construct the exemplars $E_t$, used for replays.

## 5 Results

### 5.1 RAS outperforms all the CSEGG baselines in Scenarios 1 and 2

The results for Avg. R, F, mR, mF, FWT, and BWT in learning scenarios 1 (S1) and 2 (S2) are presented in **Tab. 2**. Our observations align with established research in continual learning, especially in image classification and object detection: regardless of the SGG architectures, over both learning scenarios, Naive consistently performs the worst, showcasing significant catastrophic forgetting.

| | SGTR[32] | | | | | | | | | | | |
|---|---|---|---|---|---|---|---|---|---|---|---|---|
| Methods | Learning Scenario 1 (S1) | | | | | | Learning Scenario 2 (S2) | | | | | |
| | Avg.R ↑ | F↑ | mR↑ | mF↑ | FWT↑ | BWT↑ | Avg.R↑ | F↑ | mR↑ | mF↑ | FWT↑ | BWT↑ |
| Joint | 20.15 | 0 | 4.6 | 0 | - | - | 12.64 | 0 | 9.84 | 0 | - | - |
| Replay@100% | 16.17 | -12.24 | 3.32 | -1.34 | -1.77 | -11.72 | 4.56 | -4.13 | 4.56 | -5.61 | -1.045 | -30.25 |
| Naive | 1.33 | -28.7 | 0.86 | -1.74 | -2.03 | -60.67 | 0.51 | -23.22 | 0.05 | -11.31 | -3.77 | -62.34 |
| EWC[24] | 1.89 | -28.4 | 0.96 | -1.72 | -1.17 | -52.45 | 0 | -23.22 | 0 | -11.31 | -2.65 | -50.12 |
| RAS_GT | 5.78 | -26.51 | 1.43 | -1.54 | -1.2 | -44.27 | 0.98 | -23.11 | 0.76 | -10.86 | -1.6 | -43.25 |
| PackNet[44] | 7.19 | -25.67 | 1.35 | -1.64 | -1.03 | -42.35 | 1.67 | -22.77 | 0.9 | -10.33 | -1.4 | -42.45 |
| Replay@10% | 8.55 | -22.21 | 4.33 | -1.44 | **4.29** | -38.35 | 1.81 | -20.72 | 1.15 | -9.64 | -0.9 | -40.67 |
| Replay@20% | 9.25 | -20.35 | 4.78 | -1.42 | 3.21 | -31.98 | 2.57 | -17.17 | 1.56 | -8.07 | -0.67 | -38.27 |
| **Ours*** | **10.78** | **-18.92** | **5.6** | **-1.39** | 2.3 | **-25.56** | **3.45** | **-10.23** | **2.75** | **-6.57** | **-0.54** | **-35.67** |
| | TCNN[72] | | | | | | | | | | | |
| Methods | Learning Scenario 1 (S1) | | | | | | Learning Scenario 2 (S2) | | | | | |
| | Avg.R↑ | F↑ | mR↑ | mF↑ | FWT↑ | BWT↑ | Avg.R↑ | F↑ | mR↑ | mF↑ | FWT↑ | BWT↑ |
| Joint | 19.53 | 0 | 3.9 | 0 | - | - | 4.3 | 0 | 3.7 | 0 | - | - |
| Replay@100% | 13.45 | -8.83 | 3.6 | -0.35 | -1.5 | -10.45 | 12.45 | -4.13 | 3.2 | -0.56 | -2.1 | -20.34 |
| Naive | 0.98 | -21.2 | 0.74 | -1.35 | -3.45 | -43.87 | 0 | -18.22 | 0.45 | -2.67 | -4.12 | -53.12 |
| EWC[24] | 2.36 | -21.05 | 0.67 | -1.34 | -2.34 | -39.89 | 0 | -18.22 | 0.03 | 0 | -3.77 | -51.67 |
| PackNet[44] | 3.2 | -19.7 | 1.1 | -1.13 | -1.3 | -32.45 | 1.1 | -17.82 | 0.84 | -1.97 | -2.84 | -40.34 |
| Replay@10% | 5.67 | -18.9 | 3.21 | -1.05 | **1.45** | -28.34 | 1.81 | -16.72 | 1.03 | -1.74 | -1.4 | -43.56 |
| Replay@20% | 6.23 | -17.45 | 3.5 | -1.01 | 1.01 | -24.32 | 2.37 | -15.17 | 1.45 | -1.53 | -1.1 | -38.56 |
| **Ours*** | **7.8** | **-15.67** | **3.9** | **-0.95** | 0.5 | **-19.83** | **4.67** | **-11.31** | **2.2** | **-0.89** | **-0.97** | **-29.65** |

Table 2: **Results of CSEGG for various continual learning methods applied on the two SGG backbones (SGTR and TCNN) in Learning Scenarios 1 and 2.** See **Sec. 3.2** for continual learning baselines. See **Sec. 3.3** for evaluation metrics. The higher the evaluation metrics, the better. The best are in bold. * means the experiment is still running, we will report the results in the final version.

| Model | mR↑ | mF↑ |
|---|---|---|
| LVIS@Replay@10% | 3.98 | -1.54 |
| BLS@Replay@10% | 4.34 | -1.47 |
| **RAS (ours)** | **5.6** | **-1.39** |

Table 3: **Results at Task 5 in Learning Scenario 1 when sampling techniques are applied to long-tailed distribution data.** See **Sec. 3.2** for the sampling techniques on long-tailed distributions. We copy the results of our RAS from **Tab. 2** for easy comparisons. The best results are in bold.

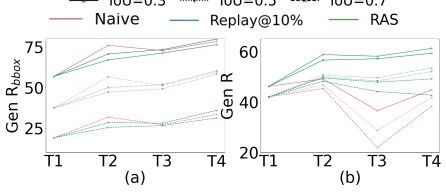

Figure 4: **Results in Scenario 3.** See **Sec. 3.3** for evaluation metrics. The higher the values, the better. Line colors indicate continual learning methods. Line types denote the IoU thresholds.

Replay-based methods, such as Replay@10% and Replay@20%, outperform techniques like EWC and PackNet. However, none of them surpass our RAS.

RAS achieves superior performance compared to Replay@20% (∼2 Gb) while requiring less storage (∼1.2 Gb), equivalent to storing 15% exemplary images. This storage efficiency is due to only needing to store the old model snapshot, triplet labels in $U_t$, and the image generation model, thus avoiding privacy concerns. Also, RAS outperforms RAS_GT (**Tab. 2**), indicating that decomposing scene graphs into smaller, more diverse ones, along with more comprehensible prompts, is more effective than storing the ground truth scene graphs and directly using them for image generation.

In Learning Scenario 2 (S2), the task involves classifying both new objects and new relationships, significantly escalating the level of difficulty compared to S1. As evident from **Tab. 2**, all CSEGG models, including Replay@100%, exhibit a decline in overall performance when compared to the upper bound Joint. Although our RAS achieves the leading performance among all the baselines, its performance is still far from Joint. This underscores the persistent challenge posed by S2 in the context of CSEGG. Future work should explore new approaches to address this gap.

To tackle the issue of imbalanced data distribution in real-world scenarios, we incorporate two established data sampling techniques (LVIS@Replay@10% and BLS@Replay@10%) into our experiment (**Sec. 3.2**). The outcomes in learning scenario S1 are presented in **Tab. 3**. We observed that their performance falls short of our RAS, underscoring the efficacy of RAS in addressing long-tailed distribution during generative replays. We also noticed that BLS@Replay@10% significantly

|  | $\gamma$ | Context | LTD | Triplet | Avg.R ↑ | F ↑ | mR ↑ | mF ↑ |
|---|---|---|---|---|---|---|---|---|
| A1 | 10 | ✗ | ✓ | Multiple | 8.23 | -21.35 | 3.98 | -1.98 |
| A2 | 10 | ✓ | ✗ | Multiple | 9.75 | -20.65 | 4.12 | -1.65 |
| A3 | 10 | ✓ | ✓ | Single | 7.75 | -22.45 | 3.12 | -2.48 |
| A4 | 2 | ✓ | ✓ | Multiple | 2.45 | -27.43 | 0.45 | -9.84 |
| A5 | 4 | ✓ | ✓ | Multiple | 5.67 | -26.42 | 2.41 | -3.26 |
| A6 | 8 | ✓ | ✓ | Multiple | 7.89 | -22.42 | 3.89 | -2.17 |
| **Ours** | 10 | ✓ | ✓ | Multiple | **10.78** | **-18.92** | **5.6** | **-1.39** |

Table 4: **Ablation results of our RAS on learning scenario S1 reveals key design insights.** This table presents the results of ablation studies conducted to identify key components of our method, as discussed in **Sec. 5.3**. Results in Avg.R, F, mR, mF are reported after the last task in Scenario S1.

outperforms LVIS@Replay@10%, contrary to findings in the classical SGG problem where BLS is considered more effective than LVIS [33]. The performance difference may stem from variations in the number of replay instances between the two approaches after applying these data re-sampling methods to exemplar images in the memory buffer (**Sec. 3.2**). This observation suggests that the original sampling methods designed for addressing long-tailed distributions in the classical SGG problem may not be as effective when applied to CSEGG.

We explored the impact of task sequence permutations on CSEGG performance, finding an effect consistent with existing literature [63] (**Fig. S6**; **Sec. A.6.1**). We also observed that fine-tuning DETR in S1 has minimal impact on forgetting, indicating that any forgetfulness in S1 is solely due to relationship incremental learning (**Fig. S7**; **Sec. A.6.2**). Moreover, to gain qualitative insights, we provide visualizations of predicted scene graphs for all CSEGG baselines in Scenario 1 (**Fig. S8** and **Sec. A.7.1**) and Scenario 2 (**Fig. S9** and **Sec. A.7.2**).

## 5.2 CSEGG Models Can Generalize in Unknown Scenes

**Fig. 4** illustrates the generalization results for detecting unknown objects and classifying known relationships among these objects in Learning Scenario 3 (S3). In **Fig. 4 (a)**, an increasing trend in Gen $R_{bbox}$ is observed with the increasing task number for all CSEGG methods, indicating improved generalization in detecting unknown objects. Notably, even with minimal training in Task 1, all CSEGG methods propose 23% reasonable object regions with threshold IoU = 0.7, showcasing the SGTR model's ability to generalize to locate "objectness". As expected, with an increase in IoU threshold from 0.3 to 0.7, we found that Gen $R_{bbox}$ decreases due to fewer bounding boxes being considered correct. Moreover, we also compared the generalization performance in object detection between Replay@10% and Naive. Contrary to previous observations in S1 and S2 (**Tab. 2**), we found that Replay@10% show a decline in Gen $R_{bbox}$, possibly due to a fixed number of detected object bounding boxes output by CSEGG methods. Similarly, our RAS exhibits a reduced G $R_{bbox}$ compared to Replay@10% and Naive, likely for the same underlying reason.

In **Fig. 4 (b)**, Replay@10% outperforms Naive in Gen R when considering correctly detected unknown object locations, emphasizing that minimizing forgetting in continual learning enhances the SGTR model's overall relationship generalization in unknown scene understanding. However, the performance of Replay@10% is still inferior to our RAS method; implying that our RAS is more proficient in generalizing to classify relationships among unknown objects. Interestingly, we also noted that even with minimal training in Task 1, all the CSEGG methods achieve 45% recall of known relationships among unknown objects, demonstrating the SGTR model's ability to generalize to classify "relationship". Visualization examples, when CSEGG models can generalize to recognize relationships, are presented in **Fig. S10** and **Sec. A.7.3**.

## 5.3 Ablation Studies on Our RAS Reveal Key Design Insights

We introduce our default method designs in **Sec. 4**. Here, we vary the components in our RAS to reveal key design insights. We propose a context checker in RAS. Here, we conduct an ablation by removing this module. Triplet labels are randomly selected and combined for text prompts. In A1 of **Tab. 4**, we observe a performance decrease of approximately 2% across all evaluation metrics, compared with our RAS. This suggests that generating images adhering to real-world context rules is crucial for replays. The lower performance may be attributed to the challenge of generating

good-quality out-of-context images for Stable Diffusion Models and the potential domain differences affecting the SGG model $M_t$ in predicting out-of-context SGG notations.

The LTD sampling module in our RAS is designed to balance the distribution of head and tail triplet labels from $U_t$. Here, we remove the LTD sampling module and report the performance of the ablated method in A2 of **Tab. 4**. Compared to our RAS, we observe an absolute decrease of 1-2% across all metrics. Notably, the relative decrease is more pronounced in mR and mF than Avg.R and F. As mR and mF indicate mean Recall and mean Forgetfulness over both tail and head classes, the larger drops in these metrics suggest that the absence of LTD sampling significantly hinders the SGG model's ability to predict tail classes from previous tasks.

In our RAS, we employ multiple triplet labels to construct text prompts for image generation. In contrast to single triplet labels, our approach yields rich text descriptions of complex scenes, allowing the SGG model to capture intricate relationships among multiple objects in the same scene. Additionally, using multiple triplets is more efficient in rehearsing, as it enables the model to practice predicting multiple triplets simultaneously within the same number of synthesized images. Indeed, when we replace multiple triplet labels with single triplet labels for text prompts, we note a decrease of approximately 3% across all metrics (compare A3 with ours in **Tab. 4**).

Lastly, we investigate the impact of generating $\gamma$ images using the same text prompt in RAS, varying $\gamma$ from 2 to 8 (**Tab. 4**, A4-6). As expected, performance improves with higher $\gamma$, showing that increased sample diversity enhances CSEGG performance. With ample computing resources, dynamically synthesizing more images could further improve performance. This highlights RAS's advantage in generating numerous images for replays without expanding storage usage.

# 6 Discussion

In the dynamic world, adapting scene graph generation (SGG) models to new objects and relationships poses challenges. Despite progress in SGG and continual learning, there is still a gap in understanding Continual Scene Graph Generation (CSEGG). We address this by operationalizing CSEGG, and introducing benchmarks, datasets, and evaluation protocols. Our study explores three learning scenarios, analyzing continual object detection and relationship classification in long-tailed class-incremental settings for CSEGG baselines. Our findings show that integrating sampling methods with CSEGG baselines to address long-tailed distributions moderately eliminates forgetting; however, a large performance gap between current CSEGG baselines and the joint training upper bound persists. To address CSEGG challenges, we propose RAS, a Replays via Analysis by Synthesis method. RAS parses previous task scene graphs into triplet labels for diverse in-context scene graph reconstruction. Based on these re-compositional context-congruent scene graphs, RAS synthesizes images with Stable Diffusion models for replays. Unlike other image replay methods, RAS stores only triplet labels and the model snapshot, maintaining constant memory usage and preserving privacy. Extensive experiments demonstrate our RAS's superior performance over current CSEGG baselines in knowledge transfers and reducing forgetting. Interestingly, our RAS model is also capable of generalizing to classify known relationships among unseen objects.

Moving forward, there are several key avenues for future research. First, our current endeavors focus on tackling CSEGG problems from static images in an Independent and Identically Distributed (i.d.d) manner, diverging from how humans learn from video streams. Future research can look into CSEGG problems on video SGG datasets. Second, our plans also involve expanding the set of continual learning baselines and integrating more long-tailed distribution sampling techniques. Third, we aim to construct a synthetic SGG dataset to systematically quantify the aspects of SGG that influence continual learning performance under controlled conditions. In RAS, SGG annotations for synthesized images in the replay buffer are predicted by the preceding SGG model, which can lead to error propagation across training iterations. In the future work, integrating a generative model with fine-grained control signals (such as bounding boxes and captions) [36] could provide more precise supervision, potentially mitigating these accumulated errors and further enhancing the performance of our approach. Although the CSEGG method holds promise for many downstream applications like monitoring systems, medical imaging, and autonomous navigation, we should also be aware of its misuse in privacy, data biases, fairness, security concerns, and misinterpretation (see **Sec. A.8** for an expanded discussion). We invite the research community to join us in expanding and updating the safe use of CSEGG benchmarks, thereby fostering its advancements in research and technology.

## Acknowledgement

This research is supported by the National Research Foundation, Singapore under its AI Singapore Programme (AISG Award No: AISG2-RP-2021-025), and its NRFF award NRF-NRFF15-2023-0001. We also acknowledge Mengmi Zhang's Startup Grant from Agency for Science, Technology, and Research (A*STAR), Startup Grant from Nanyang Technological University, and Early Career Investigatorship from Center for Frontier AI Research (CFAR), A*STAR.

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

# A Appendix

## A.1 Introduction to Three Learning Scenarios

Within this section, we present more details of three learning scenarios and their practical applications.

### A.1.1 Scenario 1 (S1): Relationship Incremental Learning

While existing continual object detection literature focuses on incrementally learning object attributes [43, 70, 8, 68, 60, 45], incremental relationship classifications are equally important as it provides a deeper and more holistic understanding of the interactions and connections between objects within a scene. To uncover contextual information and go beyond studies of object attributes, we introduce this scenario where new relationship predicates $p_k$ are incrementally added in each task (**Fig. 2 S1**). There are 5 tasks in S1. To simulate the naturalistic settings where the frequency of relationship distribution is often long-tailed, we randomly and uniformly sample relationship classes from head, body and tail categories in Visual Genome [25], and form a set of 10 relationship classes for each task. Thus, the relationships within a task are long-tailed; and the number of relationships from the head categories of each task is of the same scale. To tackle this issue, we allow CSEGG models to see the same images over tasks, but the relationship labels are only provided in their given task

The design of such images and label splits over tasks aligns with human learning scenarios where a parent teaches the baby to recognize different toys and objects in the bedroom. Though the baby is exposed to the same bedroom scenes multiple times, the parent only teaches the baby to detect and recognize one object at a time in a continual learning setting. In the future, we will expand our studies to cases where the SGG models learn from non-overlapping sets of training images for each task. The same reasoning applies in **S2** and **S3**. Example relationship classes from each task and their distributions are provided in **Fig. S1**.

Here, we provide a concrete example application of Scenario 1 in medical imaging. Within medical imaging, an agent must acquire the ability to detect cancerous cells within primary tumors, like colon adenocarcinoma. Subsequently, it must extend this proficiency to identifying the same cell types within metastatic growths that manifest in different bodily regions, such as lymph nodes or the liver. In this instance, the identical cancer cell disseminates to fresh organs or tissues, progressively establishing new relationships with other cells over the course of time.

### A.1.2 Scenario 2 (S2): Scene Incremental Learning

To simulate the real-world scenario when there are demands for detecting new objects and new relationships over time in old and new scenes, we introduce this learning scenario where new objects $O_i$ and new relationship predicates $p_k$ are incrementally introduced over tasks (**Fig. 2 S2**). To select the object and relationship classes from the original Visual Genome [25] for S2, we have two design motivations in mind. First, in real-world applications, such as robotic navigation, robots might have already learned common relationships and objects in one environment. Incremental learning only happens on less frequent relationships and objects. (2) Transformer-based AI models typically require large amounts of training data to yield good performances. Training only on a small amount of data from tail classes often leads to close-to-chance performances. Thus, we take the common objects and relationships from the head classes in Visual Genome as one task, while the remaining less frequent objects and relationships from tail classes as the other task. This results in 2 tasks in total with the first task containing 100 object classes and 40 relationship classes. In the subsequent task, the CSEGG models are trained to continuously learn to detect 25 more object classes and 5 more relationship classes. Same as **S1**, both the object class and relationship class distributions are still long-tailed within a task (**Fig. S1**).

Next, we provide two real-world example applications in robot collaborations on construction sites and video surveillance systems.

The CSEGG model's capacity to incorporate new objects and new relationships while retaining existing knowledge finds pivotal application in video surveillance contexts. Consider a company developing video-based security systems for indoor environments, capturing prevalent indoor objects and relationships. Expanding to outdoor settings like parking lots or restricted compounds demands retraining the model with new outdoor data alongside previous indoor data, ensuring operational effectiveness in both realms. The outdoor context introduces new objects like "cars" and relationships

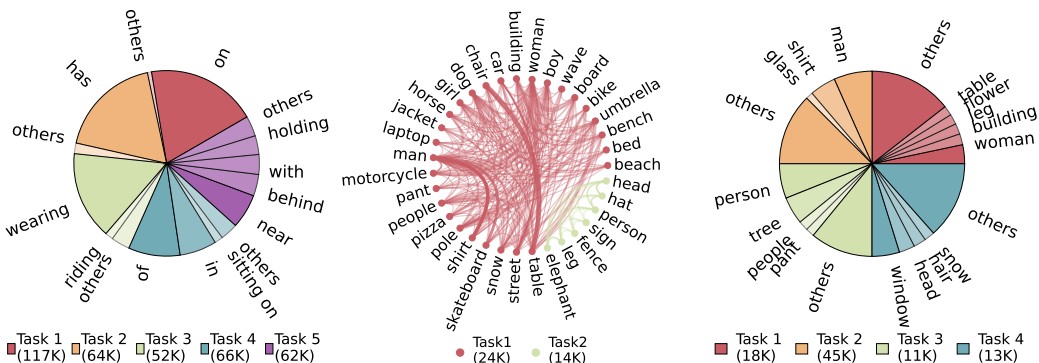

Figure S1: **Label distribution in each task in each learning scenario** is presented. In scenario S1 (a) and scenario S3 (c), we use different colors to denote different tasks. The color gradient indicates the frequency of data within a task, with the lighter color denoting the smaller frequency of data in that category. Only the most frequent labels (relationship labels in (a) and object labels in (c)) are provided. See the legend for the total data size per task. In (b) scenario S2 on both objects and relationships, data distributions are presented in the form of small-world networks, where nodes denote object categories and the edges linking object pairs indicate relationships. Thickness in edges implies the diversity of relationships between object pairs. Same color conventions as (a) and (c) are applied. See the legend for triplet sizes.

like "driving", distinct from indoor scenarios featuring "chair" and "sitting." Employing CSEGG allows the company to focus on new objects and relationships while retaining indoor insights.

Another real-world example would be a construction site where a team of robots is tasked with assembling various components to build a complex structure. Initially, during the foundation-laying phase, the robots are introduced to objects like "concrete blocks" and relationships like "stacking". As the construction advances to the wiring and installation phase, they encounter new objects like "wires" and relationships like "connecting," which were absent from earlier stages. The SGG model deployed in these robots needs to adapt incrementally to learn these new relationships without forgetting the existing ones. This ensures that the robots can effectively communicate and collaborate while comprehending the evolving scene and tasks, optimizing their construction efficiency and accuracy.

### A.1.3 Scenario 3 (S3): Scene Graph Generalization In Object Incremental Learning

We, as humans, have no problem at all recognizing the relationships of unknown objects with other nearby objects, even though we do not know the class labels of the unknown objects. This scenario is designed to investigate whether the CSEGG models can generalize as well as humans. Specifically, there are 4 tasks in total with each task containing 30 object classes and 35 relationship classes. In each subsequent task, the CSEGG models are trained to continuously learn to detect 30 more object classes and learn to classify the same set of 35 relationships among these objects. The class selection criteria for each task follow the same as **S1**, where the selections occur uniformly over head, body, and tail classes. Example object classes and their label distributions for each task are provided in **Fig. S1**. Different from **S1** and **S2**, a standalone generalization test set is curated, where the objects are unknown and their classes do not overlap with any object classes in the training set but the relationships among these unknown objects are common to the training set of every task. The CSEGG models trained after every task are tested on the same generalization test sets.

Here, we provide two real-world applications of Scenario 3 in the deep sea and space explorations for autonomous navigation systems.

A prime example is the ongoing research on deep sea exploration for autonomous navigation systems, where undiscovered flora and fauna reside beneath the ocean's surface. Encountering new and unidentified species becomes manageable through SGG's ability to understand spatial relations. The robot discerns the object's proximity or orientation even without precise identification of the species, enhancing its autonomous navigation ability. Likewise, in deep space exploration, SGG aids in recognizing spatial relationships with previously unseen space debris, aiding in path-planning. In

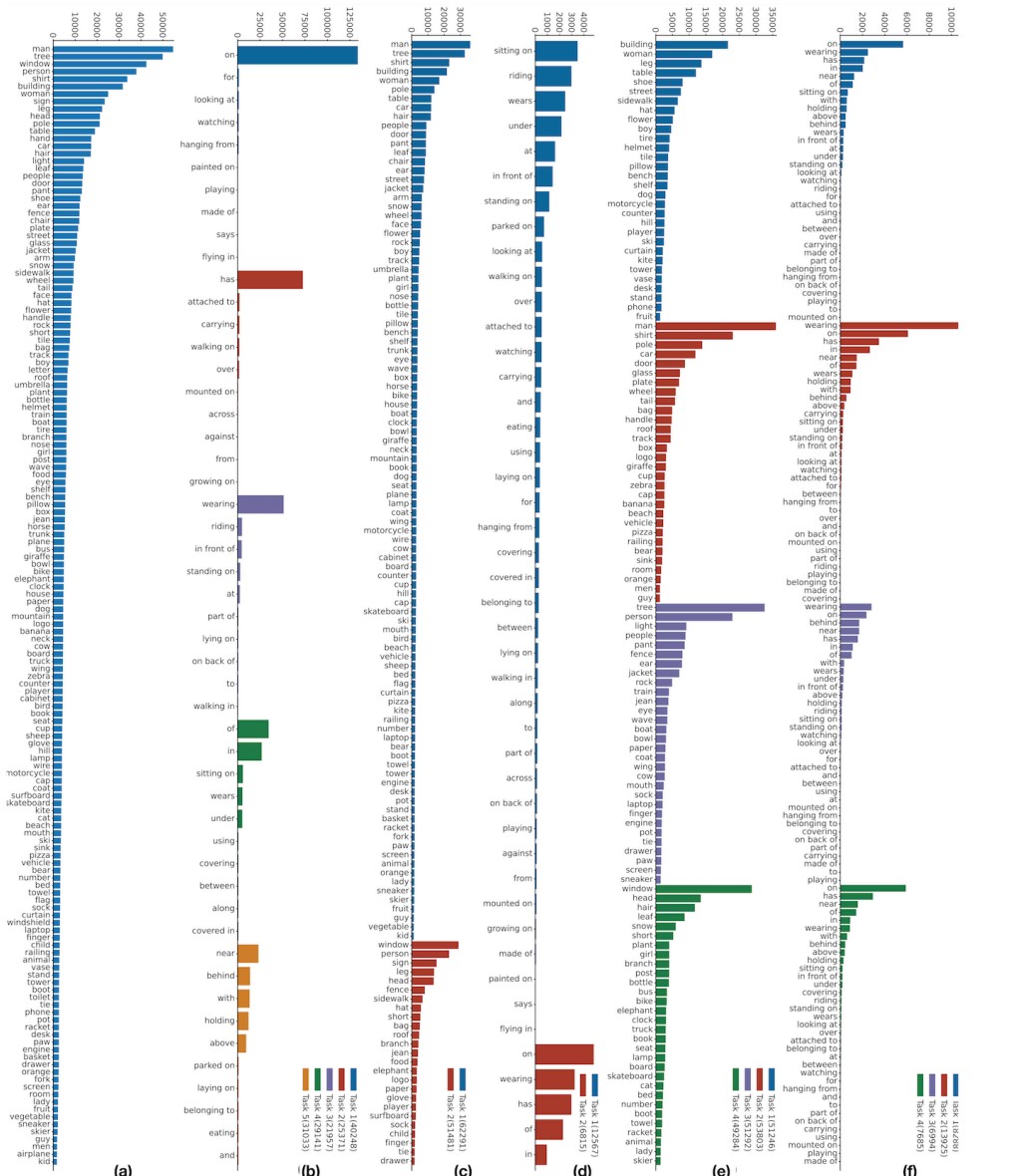

Figure S2: **Data Statistics for all Learning Scenarios**: **(a)** Distribution of objects in the entire training set of Visual Genome during Stage 1 of S1. **(b)** Distribution of relationships during Stage 2 for each task in S1. **(c)** Distribution of objects during Stage 1 for each task in S2. **(d)** Distribution of relationships during Stage 2 for each task in S2. **(e)** Distribution of objects during Stage 1 for each task in S3. **(f)** Distribution of relationships during Stage 2 for each task in S3. The numbers in brackets in the legend in **(b-f)** denote the number of training images in the particular task. Zoom in to the figure to get the exact labels and the frequency associated with them.

essence, SGG's relationship generalization empowers robots to navigate and plan routes in unfamiliar terrains, such as deep sea and deep space, where novel encounters demand adaptable responses.

## A.2    Data Statistics

In this section, we provide various types of data statistics for all three learning scenarios. Specifically, we present statistics regarding the number of images, objects, and relationships involved in each task of each learning scenario. This information is provided in **Fig. S2**.

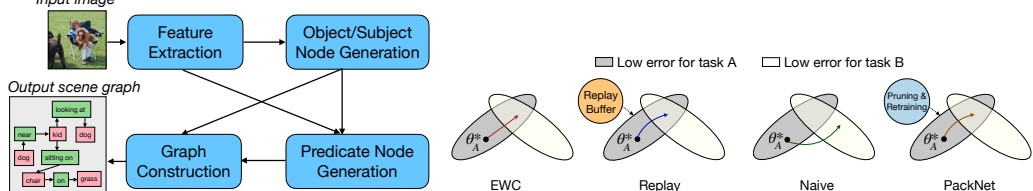

Figure S3: **Introduction to backbone SGG models and continual learning baselines.** We use Scene graph Generation TRansformer (SGTR) [32] as the SGG backbone (**Sec. 3.2**). SGTR consists of four modules indicated by each blue box. Arrows indicate the signal flows among modules. (b) Four continual learning baselines are listed: EWC [24], Replay [57], Naive (**Sec. 3.2**) and PackNet [44](**Sec. 3.2**). $\theta_A^*$ denotes the optimal network parameters after learning on task A. The arrows in colors indicate the shifts of network weights in the parameter space when learning Task B for different baselines.

## A.3 Implementation and Training Details

### A.3.1 SGTR Backbone

For SGTR in **Fig. S3 (a)**, the approach uniquely formulates the task as a bipartite graph construction problem. Starting with a scene image ($I_i$), SGTR utilizes a 2D-CNN and transformer-based encoder to extract image features. These features are then incorporated into a transformer-based decoder, predicting object and subject nodes ($O_i$). Predicate nodes ($R_i$) are formed based on both image features and object node features, and a bipartite graph ($G_i$) is constructed to represent the scene collectively. The correspondence between object nodes ($o_i$) and predicate nodes ($r_k$) is established using the Hungarian matching algorithm [26]. Experimental results are based on the average over three runs, and the implementation leverages public source codes from [32] and [68] with default hyperparameters.

The SGTR is trained in two stages in a supervised manner. In stage 1, only object detection losses in DETR is [6] applied on $O_i$. In stage 2, only predicate entity prediction loss is applied on $R_i$, which can be further decomposed into L1 and GIOU losses for object/subject/predicate localization [54] and cross-entropy loss for object/subject/predicate classification. In learning scenario S1, we skip Stage 1, and directly load pre-trained weights of DETR for object detection on the entire training set of Visual Genome [32]. In stage 2 of S1, we freeze the feature extractor, and fine-tune the rest parts of SGTR for predicate entity predictions. As only relationship classes are incrementally introduced in S1, we freeze the entire weights of DETR for detecting all the objects in the scene over tasks. However, empirical results suggest that fine-tuning transformer-based encoders in DETR helps downstream predicate predictions [32]. Even with fine-tuning DETR in S1, we verify that there is minimal forgetting of detecting all objects in the scene over tasks (see **Fig. S7** and **Sec. A.6.2**). Thus, the forgetting observed in S1 could only be attributed to incremental relationship learning. In Stage 1 of S2 and S3 where object classes are also incrementally introduced over tasks, we load weights of the feature extractor, pre-trained on ImageNet [13], and fine-tune the entire DETR [6] over all the tasks. Stage 2 of S2 and S3 is the same as S1.

Training the SGTR model involves two stages:

**Object Detection Training:** In this stage, a batch size of 32 is used. All methods are optimized using the Adam optimizer with a base learning rate of $1 \times 10^{-4}$ and a weight decay of $1 \times 10^{-4}$. Object detection training is conducted only in the S2 and S3 scenarios. Each task in S2 is trained for 100 epochs, while each task in S3 is trained for 50 epochs. To expedite convergence, pre-trained weights on ImageNet are utilized before training on Task 1 for both S2 and S3.

**SGG (Scene Graph Generation) Training:** In this stage, the entire SGTR model is fine-tuned while keeping the 2D-CNN feature extractor frozen. A batch size of 24 is employed, and the Adam optimizer is used with a base learning rate of $8 \times 10^{-5}$. In S1 and S3, each model is trained for 50 epochs per task, while in S2, 80 epochs per task are used. All models are trained on 4 A5000 GPUs.

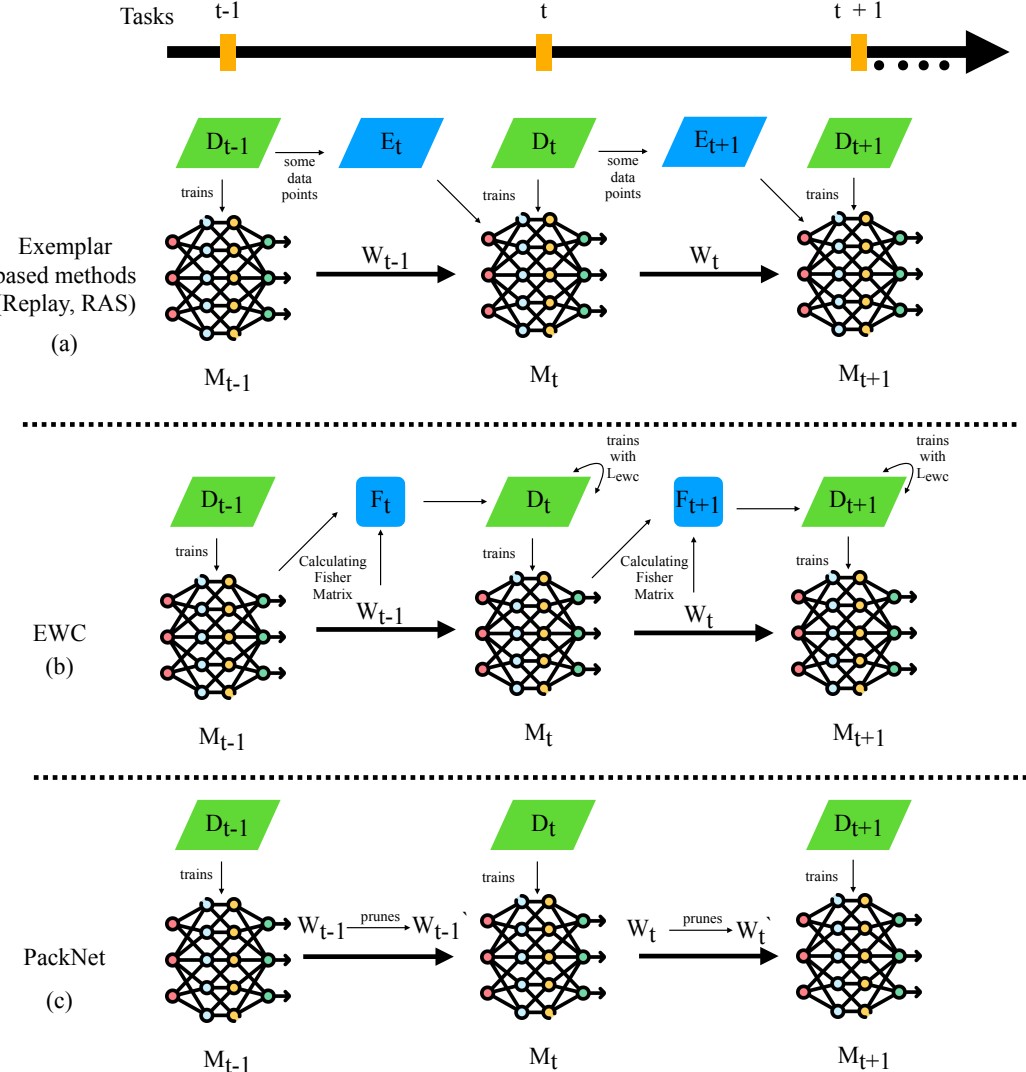

Figure S4: **Detailed Schematic of three CSEGG Baselines.** In this schematic, we show the operations of each baseline at tasks $t-1$, $t$, $t+1$. (a) shows the schematic for exemplar-based methods like replay and RAS. (b) shows the schematic for EWC. (c) shows the schematic for PackNet. Here, $D_i$ refers to the dataset of task $i$. $E_i$ denotes the exemplar set for replays at task $i$. $W_i$ is the weights of $M_i$ at task $i$.

### A.3.2 TCNN Backbone

As for TCNN, it employs Faster-RCNN [18] to generate object proposals from a scene image ($I_k$). The model extracts visual features for nodes and edges from these proposals. Through message passing, both edge and node GRUs output a structured scene graph. Experimental results are based on the average over three runs.

Given a scene image $I_i$, TCNN utilizes Faster-RCNN[18] to generate a set of object proposals. The model subsequently extracts visual features of nodes and edges from the set of object proposals. Finally, both edge and node GRUs output a structured scene graph via message passing. The TCNN is trained in two stages in a supervised manner. In stage 1, only object detection losses in Faster-RCNN[53] are applied on $O_i$. We use the cross entropy loss for the object class and $L1$ loss for the bounding box offsets. In stage 2, the visual feature extractor (VGG-16[62] pre-trained on ImageNet [13]) and GRUs layers are trained to predict the final object classes, bounding boxes, and

*"Realistic Image of man on horse and house behind horse and man in front of house."*

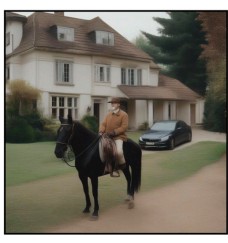

*"Realistic Image of man sitting on sofa and man sitting on chair and chair near sofa and person near man."*

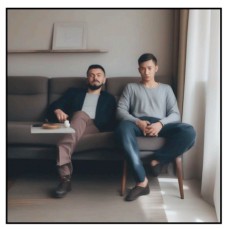

(a)                                                    (b)

Figure S5: **Visualizations of example generated images given the text prompt from our RAS.** Each row presents two examples of generated images. Given the input text prompt on the left, the generated image is displayed on the right. These visualizations provide qualitative validations into the capability of RAS to produce diverse and contextually relevant images based on our designed text prompts.

relationship predicates using cross-entropy loss and $L1$ loss. In Learning Scenario 1 (S1), similar to the implementation details of SGTR in **Sec. A.3.1**, we skip Stage 1, and directly load pre-trained weights of Faster-RCNN for object detection on the entire training set of Visual Genome [32]. In stage 2 of S1, we load the pre-trained weights of the visual feature extractor (pre-trained on ImageNet) and fine-tune the rest parts of the model. In stage 2 of S1, we load the pre-trained weights of visual feature extractor (pre-trained on ImageNet) and fine-tune the rest parts of the model. In Stage 1 of S2 and S3 where object classes are also incrementally introduced over tasks, we load weights of the Faster-RCNN, pre-trained on ImageNet [13], and fine-tune it over all the tasks. Stage 2 of S2 and S3 follows the same training regimes as Stage 2 of S1.

**Object Detection Training:** All methods are optimized using the SGD optimizer with a base learning rate of $1 \times 10^{-2}$ and a weight decay of $1 \times 10^{-4}$. For training on the entire VG dataset, we train the model for 60 epochs with a batch size of 8 for both S2 and S3. To expedite convergence, pre-trained weights on ImageNet are utilized before training on Task 1 for both S2 and S3.

**SGG (Scene Graph Generation) Training:** A batch size of 12 is employed, and the SGD optimizer is used with a base learning rate of $1 \times 10^{-2}$ and a weight decay of $1 \times 10^{-4}$ . In S1, each model is trained for 30 epochs. In S2, each model is trained for 15 epochs. In S3, each model is trained for 25 epochs. All models are trained on 4 A5000 GPUs.

## A.4 Mathematical formulations of Continual learning baselines

Here, we introduce mathematical formulations of these baselines: (1) To train Naive baseline on CSEGG, at task $t$, we take the previously trained model $M_{t-1}$ with weights $W_{t-1}$ and train it on the dataset $D_t$ at task $t$, to obtain weights $W_t$ of the model $M_t$. (2) For EWC, we use $W_t$ and $M_t$ to calculate $F_{t+1}$, where $F$ is the Fisher information matrix using the equation, $F_{t+1} = -E[\frac{\partial^2}{\partial W_t^2} log(M_t(x)|W_t)]$. During the training of task $t + 1$, we add $L_{EWC}$ to the training loss, where $L_{EWC} = \sum F_{t+1}(W_{t+1} - W_t)^2$. (3) For PackNet, after training of task $t$, we take $W_t$ and apply a pruning algorithm to obtain the pruned weights $W_t'$. At task $t+1$, we obtain $M_{t+1}$ by training $M_t$ with $W_t'$ on $D_{t+1}$. (4) For Replay@M, we create exemplar set $E_t$ at the replay buffer by storing data points from $D_{t-1}$ after training $M_{t-1}$ on task $t - 1$ . At task $t$, we obtain weights $W_t$ of $M_t$ by training $M_t$ on $E_t$ and $D_t$. (5) For the joint training, we train one model $M_T$ jointly on all the datasets $\{D_t\}$, where $t \in \{1, 2, ..., T\}$. (6) For RAS_GT, we adopt the similar math formulations as Replay@M. See more details in **Sec. 4**.

## A.5 Evaluation Metrics

To assess the catastrophic forgetting of CSEGG models, we define **Forgetfullness (F@K)** as the difference in R@K on $D_{t=1}$ between the CSEGG models trained at task $t$ and task 1. An ideal CSEGG model could maintain the same $R@K$ on $D_{t=1}$ over tasks; thus, $F = 0$ for all tasks. The more negative F is, the more severe in forgetting an model gets.

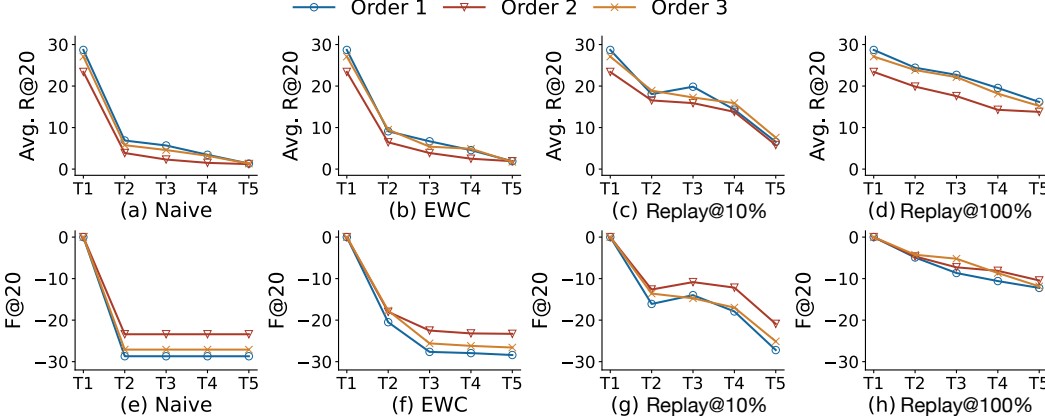

Figure S6: **Results of F@K=20, Avg. R@K=20 over tasks on CSEGG models with the SGTR backbone in Learning Scenario 1 with different permutations of task sequences.** (a),(e) denotes Avg.R@K and F@K for naive baseline. (b),(f) denotes Avg.R@K and F@K for EWC baseline. (c),(g) denotes Avg.R@K and F@K for Replay@10% baseline. (d),(h) denotes Avg.R@K and F@K for Replay@100% baseline. See **Sec. 3.2** for introduction to continual learning baselines. See **Sec. 3.3** for explanations about evaluation metrics. X-axis indicates the task numbers. The higher F, Avg.R@20, the better.

To assess the overall recall of CSEGG models over tasks, we also report the continual average recall (**Avg. R@K**). Avg. R@K is computed as the average recall on all the data at the previous and current tasks $D_i$, where $i \in \{1, 2, ..., t\}$.

To assess whether the knowledge at previous tasks facilitates learning the new task and whether the knowledge at new tasks enhances the performances at older tasks, we introduce **Forward Transfer (FWT@K)** [39] and **Backward Transfer (BWT@K)**[41]. BWT@K is defined as $BWT@K = \frac{1}{T-1}\sum_{i=1}^{T-1} R@K_{T,i} - R@K_{i,i}$, where $T$ denotes the total number of tasks in a learning scenario and $R@K_{i,j}$ denotes the continual learning model trained after task $i$ and tested in task $j$. FWT is defined as $FWT@K = \frac{1}{T-1}\sum_{i=2}^{T} R@K_{i,i} - \overline{b@K}_{i,i}$, where $\overline{b@K}_{i,i}$ is the **R@K** for an independent model with random initialization trained in task $i$ and tested in task $i$.

In learning scenario S3, we evaluate CSEGG models on their abilities to generalize to detect unknown objects and classify known relationships on these objects, in the standalone generalization test set over all tasks. To benchmark these, we introduce two evaluation metrics: the recall of the predicted bounding boxes on unknown objects (**Gen R$_{bbox}$@K**) and the recall of the predicted graph $G_i$ (**Gen R@K**). As the CSEGG models have never been taught to classify unknown objects, we discard the class labels of the bounding boxes and only evaluate the predicted box locations with **Gen R$_{bbox}$@K**. To evaluate whether the predicted box location is correct, we apply a hard threshold of Intersection over Union (**IoU**) between the predicted bounding box locations and the ground truth boxes. If the IoU exceeds the predetermined threshold, it is considered a true positive ($TP$). We used **IoU** thresholds of 0.3, 0.5, and 0.7 for our experiments. If the **IoU** exceeds the threshold multiple times for the same predicted bounding box, we consider it a single positive prediction. This is because the metric aims to define the model's performance in locating bounding boxes of objects that have not been learned during training. Counting multiple times for the same box would be misleading, as it would inflate the number of $TP$s and recall, while the actual number of unknown bounding boxes the model can generate might be low. The total possible positives ($TP + FN$) are determined by the total number of ground truth boxes in the image. In general, **Gen R$_{bbox}$@K** helps evaluate how well the CSEGG model locates unknown objects within an image. True positives ($TP$) represent successful identification of the location of an unknown object, while $TP + FN$ represents all possible unknown objects the CSEGG model could locate. Thus, object classification labels are not required to calculate **Gen R$_{bbox}$@K**. Instead, we need the total number of ground truth bounding boxes in an image and the number of predicted boxes that meet the IoU thresholds.

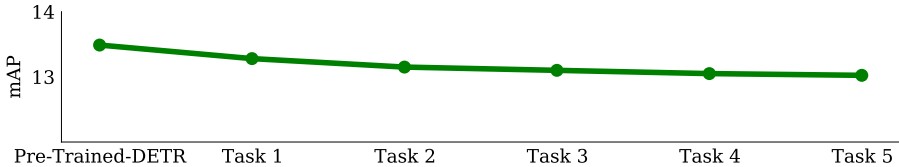

Figure S7: **mAP performance on the entire Visual Genome Test Set.** We used a pre-trained DETR checkpoint [32] as well as a naive baseline of SGTR continually trained on each task from S1, evaluated them on the entire Visual Genome's test set, and reported their mAP performances. We observed a minimal decrease in mAP across all tasks.

To assess whether the CSEGG model generalizes to detect known relationships over unknown objects, we evaluate the recall **Gen R@K** of the predicted relationships $r_k$ only on *correctly predicted* bounding boxes.

### A.6 More Result Analysis on Continual SGTR Methods

#### A.6.1 Results for Different Task Sequences

Recent work [63] has highlighted the consistent and substantial curriculum effects in class-incremental learning and continual visual question-answering tasks. Inspired by these findings, we conducted experiments to assess the impact of the curricula in the context of CSEGG. To delve into its potential influence, we trained baselines with three distinct task sequences for Learning Scenario 1. Our results demonstrated that curriculum learning indeed shapes CSEGG performance within class-incremental settings. Notably, in **Fig. S6(d)**, a large difference in Avg.R@20 between Order 1 and Order 3 emerges for the Replay (100%) baseline. Similarly, **Fig. S6(a)** reveals a substantial Avg.R@20 disparity between Order 2 and Order 3 for the Naive baseline. This trend extends to F@20, as depicted in **Fig. S6(e)(f)(g)(h)**. These insights collectively affirm the significance of the curricula within CSEGG.

#### A.6.2 Minimal Forgetting in DETR

To validate the impact of fine-tuning the DETR model in training Stage 2 of learning scenario S1 on relationship predicate predictions and to ensure minimal forgetting occurs in object detection (**Sec A.3**), we compare the mean Average Precision (mAP) for object detection on the entire test set of Visual Genome between the pre-trained DETR checkpoint from the paper [32] and the DETR models after fine-tuning on each task of S1.

As shown in **Fig. S7**, the results indicate a slight decrease of 0.4 in mAP from the pre-trained checkpoint to the DETR models over 5 tasks. This study provides evidence that fine-tuning DETR in S1 has negligible effects on forgetting. The forgetfulness observed in S1 can only be attributed to relationship incremental learning.

### A.7 Visualization Examples for All Learning Scenarios for Continual SGTR-based Models

In this section, we present visualization examples from each learning scenario to showcase the performance of the three continual SGTR-based models, namely Replay@10%, EWC, and Naive, in three learning scenarios.

#### A.7.1 Learning Scenario 1 (S1)

From **Fig.S8** we observe that, in Task 1, the ground truth scene graph contains triplets of "on" relationship: "plate on table" and "hair on women". After training on task 1, all three models (Replay@10%, EWC, Naive) can accurately predict these triplets of "on" relationship.

In Task 2, triplets of "has" relationship are introduced: "plate has food" and "women has hair". After training on task 2 data, the Replay@10% model successfully remembers the triplets of "on" relationship ("plate on table", "hair on women") from Task 1 and predicts "women has hair". The Naive model forgets the triplets of "on" relationship and only predicts "women has hair". The EWC

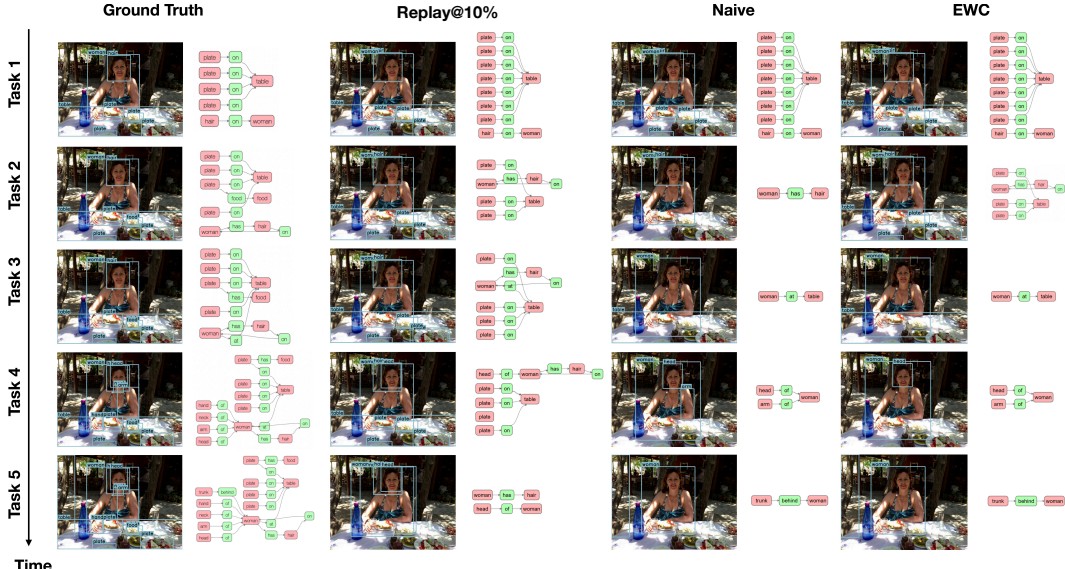

Figure S8: **Visualization example for Learning Scenario 1 (S1).** The leftmost column in the figure displays the ground truth bounding boxes and scene graphs for each task in Learning Scenario 1 (S1). The remaining columns, from left to right, represent the bounding boxes and scene graphs generated by each baseline model (Replay@10%, Naive, and EWC). In all the scene graphs, red boxes indicate objects, while green boxes represent relationships. The direction of the arrows between the red (object) and green (relationship) boxes indicates the subject and object ordering in the triplet. For example, in the scene graph predicted by the EWC model after Task 5, the triplet is "trunk behind women", as the arrow goes from "trunk" to "behind" to "women". The time arrow on the left side of the figure demonstrates that the model is exposed to new data over time, with new relationships incrementally added, as described in **Sec. 3.1**.

model remembers the triplets of "on" relationships and predicts "women has hair". None of the models predict "plate has food".

In Task 3, triplet of "at" relationship is introduced: "women at table". After training on task 3 data, the Replay@10% model remembers the previous triplets ("plate on table", "women has hair", "hair on women") and predicts "women at table". The Naive model forgets the previous triplets and only predicts "women at table". In contrast to its previous performance, the EWC model forgets the previous triplets and only predicts "women at table".

In Task 4, triplets related to "of" relationship are introduced: "hand of women", "neck of women", "arm of women", and "head of women". After training on task 4 data, the Replay@10% model remembers the triplets related to "on" and "has" relationships ("plate on table", "women has hair", "hair on women") from previous tasks but forgets the "at" relationship triplet ("women at table"). It only predicts "head of women" from the triplets introduced in Task 4. The Naive and EWC models both forget the "at" relationship triplet from the previous task but predict "head of women" and "arm of women" from the triplets introduced in Task 4.

In Task 5, triplet belonging to the "behind" relationship is introduced: "trunk behind women". After training on task 5 data, the Replay@10% model forgets the triplets related to "on" relationship ("plate on table", "hair on women") and only remembers the triplets related to "has" and "of" relationships ("women has hair", "head of women") learned from the previous task. It is not able to predict "trunk behind women". The Naive model, similar to its performance after previous tasks, fails to remember any triplets previously learned and only predicts "trunk behind women". The EWC model also fails to remember any triplets from the previous task and only predicts "trunk behind women".

### A.7.2 Learning Scenario 2 (S2)

From **Fig.S9**, we observe that, in Task 1, the ground truth scene graph contains triplets: "man riding skateboard", "man above skateboard", and "shoe of skateboard". After training on task 1 data,

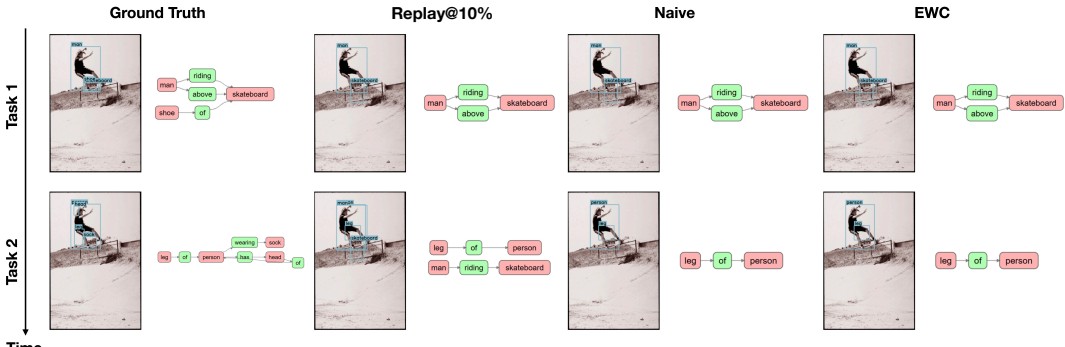

Figure S9: **Visualization example for Learning Scenario 2 (S2).** The leftmost column shows the ground truth bounding boxes and scene graphs in each task of Learning Scenario 2 (S2). The remaining columns, from left to right, represent the bounding boxes and scene graphs generated by each baseline model (Replay@10%, Naive, and EWC). In all the scene graphs, red boxes indicate objects, while green boxes represent relationships. As explained in **Fig. S8** caption, the direction of the arrows between the red (object) and green (relationship) boxes indicates the subject and object ordering in the triplet. The time arrow on the left side of the figure demonstrates that the model is exposed to new data over time, with new objects and relationships incrementally added, as described in **Sec. 3.1**.

all three models (Replay@10%, Naive, EWC) predict "man riding skateboard" and "man above skateboard". None of the models predict "shoe of skateboard".

In Task 2, new triplets introduced are: "leg of person", "person wearing sock", "person has head", and "head of person". After training on task 2 data, the Replay@10% model only remembers "man riding skateboard" from the previous task, forgetting "man above skateboard". Moreover, Replay@10% model can only predict "leg of person" from the triplets introduced in task 2. The Naive model forgets all the triplets from task 1 ( "man riding skateboard", "man above skateboard") and only predicts "leg of person" from the triplets introduced in task 2. Similar to Naive model, EWC model forgets all the triplets from task 1 ( "man riding skateboard", "man above skateboard") and only predicts "leg of person" from the triplets introduced in task 2.

### A.7.3    Learning Scenario 3 (S3)

**Fig. S10** illustrates the performance of the Replay@10% and Naive models in locating unknown objects and recognizing the relationships between these objects and other nearby unknown objects. The ground truth in **Fig. S10** consists of three unknown objects: "mountain", "sheep", and "house", along with three relationships: "near", "behind", and "infront of" (between "mountain" and "house"), and a "near" relationship (between "sheep" and "house").

After training on Task 1 data, the Naive model can accurately locate the objects "mountain" and "house". However, no new object, such as "sheep", is located by the Naive model after training on Task 2 data. After training on Task 3 data, the Naive model is also able to locate the object "sheep" in addition to "mountain" and "house". Even after training on Task 4 data, the Naive model continues to locate all three objects: "mountain", "house", and "sheep". In contrast, the Replay@10% model, after training on Task 1 data, can only locate "mountain" and "house". This remains the same even after training on Task 2 and Task 3 data, where the Replay@10% model can still only locate the objects "mountain" and "house". However, after training on Task 4 data, the Replay@10% model is able to locate all the objects: "mountain", "house", and "sheep".

Regarding relationship generalization on unknown objects, the Naive model, after training on Task 1, can only predict the "near" relationship between the located objects "mountain" and "house" out of the three possible relationships. This performance remains the same even after Task 2. However, after training on Task 3, the Naive model can predict the "behind" relationship in addition to the "near" relationship between the located objects "mountain" and "house". After Task 4, the Naive model can predict the "behind" and "near" relationships between the located objects "mountain" and "house", as well as the "near" relationship between the located objects "sheep" and "house".

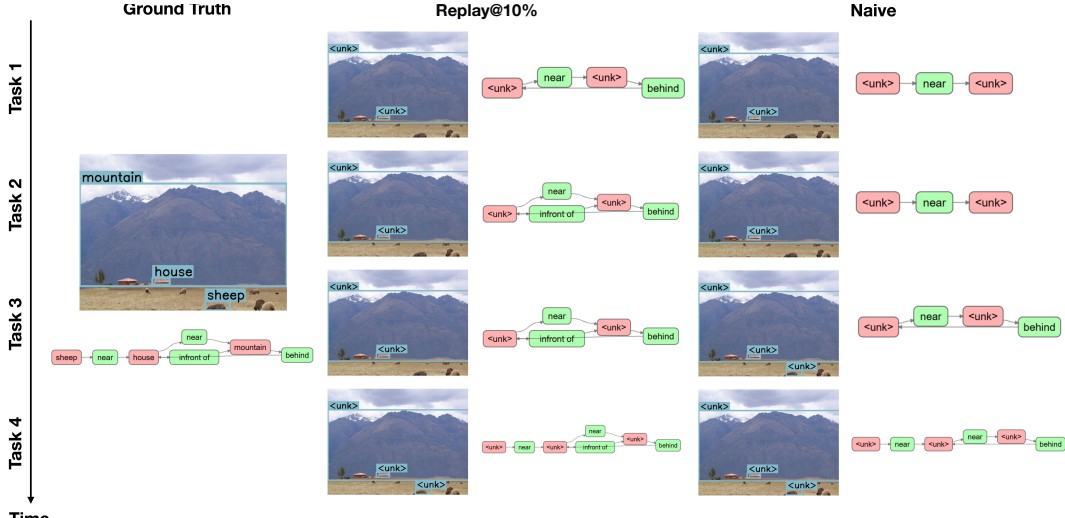

Figure S10: **Visualization example for Learning Scenario 3 (S3).** The leftmost column shows the standalone ground truth bounding boxes and scene graphs in the generalization test set regardless of which task it is in Learning Scenario 3 (S3). The remaining columns, from left to right, represent the bounding boxes and scene graphs generated by each baseline model (Naive, Replay@10%). Similar to **Fig. S8**, and **S9**, the red boxes in all scene graphs indicate objects , while green boxes represent relationships. As explained in **Fig. S8** caption, the direction of the arrows between the red (object) and green (relationship) boxes indicates the subject and object ordering in the triplet. The time arrow on the left side of the figure demonstrates that the model is exposed to new objects over time as described in **Sec. 3.1**. For easy referral of object instances in the predicted scene graphs, we numbered the unknown bounding boxes in this figure, where the numbers are not actually present in the model predictions. Concretely, unk_1 refers to "mountain"; unk_2 is "house"; and unk_3 is "sheep".

In contrast, the Replay@10% model, after training on Task 1, can predict the "near" and "behind" relationships between the located objects "mountain" and "house". After Task 2, it can also predict the "infront of" relationship between the located objects "mountain" and "house" along with "near" and "behind" relationships. Even after Task 3, the Replay@10% model is still able to predict the "near", "behind", and "infront of" relationships between the located objects "mountain" and "house". After Task 4, as it can now locate the object "sheep", the Replay@10% model can also predict the "near" relationship between the objects "house" and "sheep", in addition to the existing relationships between "mountain" and "house".

## A.8   Ethical Concerns

The development and deployment of Scene Graph Generation (SGG) technology present potential negative societal impacts that warrant careful consideration [32]. Firstly, privacy concerns arise as SGG may inadvertently capture sensitive information from images, potentially violating privacy rights and raising surveillance issues. Secondly, bias and fairness challenges persist, as SGG algorithms can perpetuate biases present in training data, leading to discriminatory outcomes that reinforce societal inequalities. Misinterpretation and misclassification by SGG algorithms could result in misinformation and incorrect actions, impacting decision-making. The risk of manipulation and misuse of SGG-generated scene representations for malicious purposes is also a concern. For example, attackers might manipulate scene graphs to deceive systems or disrupt applications that rely on scene understanding.

