# OpenReview forum: "Adaptive Visual Scene Understanding: Incremental Scene Graph Generation"
_NeurIPS.cc/2024/Conference — NeurIPS 2024 poster_

### Official Review · Reviewer_XGhv · 2024-07-02

**Soundness:** 2
**Presentation:** 2
**Contribution:** 2
**Rating:** 3
**Confidence:** 4

**Summary:**

This paper proposes a new task, Continual Scene Graph Generation (CSEGG), where SGG models must dynamically update to recognize new objects and relationships. It introduces a benchmark with three learning regimes: relationship incremental, scene incremental, and relationship generalization. In addition, the paper proposes a learning method called Replays via Analysis by Synthesis (RAS), which utilizes textual triplets to synthesize new images for model training. Performance evaluation on a dataset shows the effectiveness.

**Strengths:**

1. The paper is clear and easy-to-understand.
2. The effectiveness of key components has been verified.

**Weaknesses:**

1. The significance of CSEGG is limited. In CSEGG, the model should be trained for new categories. However, such training processes are unstable, considering the data, hyperparameters, and hardware requirements. What are the advantages of CSEGG compared to open-vocabulary SGG [a] and zero-shot SGG [b]?
2. In RAS, the training targets are predicted by the previous SGG model, which may cause error propagation. Why not use a generative model [c] with fine-grained control signals (e.g., given boxes and captions)?
3. The two-stage SGG baseline (IMP, CVPR'17) is very old. Why not use the latest models (e.g., PE-Net [d])?
4. The model is evaluated on VG only. More experiments are needed to verify the generality, e.g., cross-domain evaluation.



[a] Expanding Scene Graph Boundaries: Fully Open-vocabulary Scene Graph Generation via Visual-Concept Alignment and Retention. arXiv, 2023.
[b] Zero-shot Visual Relation Detection via Composite Visual Cues from Large Language Models. NeurIPS, 2023.
[c] Open-Set Grounded Text-to-Image Generation. CVPR, 2023.
[d] Prototype-based embedding network for scene graph generation. CVPR, 2023.

**Questions:**

1. What does the "long-tailed distribution" in L43 refer to? long-tailed predicate distribution or long-tailed object distribution?

**Limitations:**

The authors discussed the limitations in the appendix.

---

> ### Author Rebuttal · Authors · 2024-08-07
>
> **[XGhv.Weakness.1 - CSEGG is limited]** We would like to address the concerns regarding the difference of CSEGG compared to open-vocabulary SGG [a] and zero-shot SGG [b].
>
> First, we respectfully disagree with the reviewer that the significance of CSEGG is limited. The problem setting in open-vocabulary SGG [a] and zero-shot SGG [b] differs fundamentally from the problem setting addressed by CSEGG. In CSEGG, we aim to simulate scenarios where the “new” predicates or “new” objects encountered by the model are completely novel and have not been seen by any models before including the existing language models or models with multiple modalities.
>
> For example, if a new drug called XYZ is discovered, this drug has not been introduced to any AI models, whether they are language, vision, or multi-modal models. As a result, all the zero-shot and open-vocabulary models referenced in the two citations would fail to identify this new drug, as there is NO prior knowledge about this drug to transfer knowledge at the first place.
>
> Second, open-vocabulary SGG [a] uses a frozen text encoder with pre-learned weights, assuming prior knowledge of objects or predicates. Similarly, zero-shot SGG [b] relies on large language models for description-based prompts, assuming these models have encountered the new objects or predicates. In contrast, CSEGG simulates scenarios where completely novel objects or predicates are introduced. Relying on pre-learned information from large language models or frozen text encoders would violate the continual learning setting.
>
> Both [a] and [b] are fantastic works. We will cite them and discuss the differences of these works from our work in the related work section. However, our work focuses on continual learning in SGG, which is NOT addressed by these methods as it is a different problem setting altogether.
>
>
> **[XGhv.Weakness.2 - Fine-grained controls for RAS]** We appreciate the reviewer’s suggestion. However, we would like to point out that even without fine-grained controls as suggested in [c], our approach has already demonstrated decent performance outperforming all the existing continual learning baselines.
>
> As we are the first to address continual learning in SGG, we introduced a proof-of-concept model called RAS. Our main goal was to show that generative modeling can prevent catastrophic forgetting in CSEGG.
>
> We agree that using generative models with fine-grained control signals like boxes and captions could improve our method. We will cite the suggested work in the related works section and incorporate it in our model in the final version.
>
> **[XGhv.Weakness.3 - Limited Choice of SGG Models]** We understand the reviewer's concern that the two-stage SGG baseline (IMP, CVPR'17) is old. However, with 1359 citations, it remains a pioneering model in SGG and a strong baseline for comparison in subsequent works. We included it to show that our method works on classical SGG models and that the CSEGG benchmark can be extended to other SGG models. Additionally, we included a one-stage baseline (SGTR, CVPR'22) to validate our benchmark with more recent models.
>
> We agree that PE-Net [d] is a very interesting work, and we plan to try and include this in the CSEGG benchmark for the final version.
>
> We included only one model from each category (one-stage and two-stage) due to hardware and time constraints. For Learning Scenario 1 (S1), we conducted 120 experiments per model, including 5 tasks for each of the 8 baselines (excluding joint models), with 3 runs per baseline for statistical significance. For Learning Scenario 2 (S2), we performed 48 experiments per model, involving 2 tasks for each of the 8 baselines, also with 3 runs per baseline. This totals 168 experiments per SGG model, and with 2 models, we conducted 336 experiments. Each experiment takes about 2 days on a single machine with 4 A5000 GPUs. Using 3 machines, training for S1 and S2 alone took approximately 7 months. Including Learning Scenario 3 (S3) and additional ablation studies, the entire set of experiments extended to around 9-10 months. Thus, we included only one model from each category (one-stage and two-stage) in our study. We invite the community to contribute by incorporating more SGG backbones.
>
> **[XGhv.Weakness.4 - No generality]** We agree with the reviewer that studying model generalization in scene graph generation is interesting. This includes training a SGG model in VG dataset and testing on other SGG datasets, such as OpenImages. However, we argue that model generalization and continual learning are completely different problems. In our work, we focus on continual learning.
>
> Ideally, we would like to include other SGG datasets such as OpenImages, for our Continual SGG problem. However, our current analysis is limited to the Visual Genome (VG) dataset due to hardware restrictions and time constraints as mentioned in **[XGhv.Weakness.3]**.
>
> We appreciate the list of references related to open-vocabulary, open-set, and zero-shot problems in SGG. However, we would like to emphasize again that none of these works look into the problem of continual learning, which is the main focus of our work. We will cite these papers and discuss the differences from our work in the related work section. As mentioned in **[XGhv.Weakness.1]**, our work studies continual learning problems in SGG, which is valuable, unique, and challenging. It is different from open-vocabulary, open-set, and zero-shot SGG.
>
> **[XGhv.Question.1 - "long-tailed distribution" in L43]** We refer “long-tailed distribution” to distributions in both objects and relationships; and hence, predicates altogether. We will clarify this statement in the final version.  We present histograms of object and relationship distributions in each task of each learning scenario in Fig. 5 in the Appendix. The histograms show that the long-tailed distribution of objects and relationships (predicates) persists across all tasks and scenarios.

---

> > ### Comment · Reviewer_XGhv · 2024-08-11
> >
> > Thanks for the rebuttal. I read the other reviewers' comments and the rebuttal. Overall, I am inclined to reject this paper.
> >
> > I want to mention that, the description-based paradigm does NOT assume the model has encountered new objects or predicates. LLM can generate descriptions for any object/predicate categories, and the generated descriptions can be used by the text encoder of VLMs. See [a,b] for more details.
> >
> > [a] Visual Classification via Description from Large Language Models, ICLR 2022.
> > [b] Zero-shot Visual Relation Detection via Composite Visual Cues from Large Language Models. NeurIPS, 2023.

---

> > > ### Author Response · Authors · 2024-08-11
> > > **Language models are NOT equipped with continual learning ability without forgetting; zero-shot is NOT equal to continual learning**
> > >
> > > We thank the reviewer for the response. However, we respectfully disagree with the reviewer that our work lacks contributions because we did not use language models to tackle the problem of zero-shot. We are interested in continual learning problems in scene understanding in vision. Just like humans, we encounter new objects/relationships/contextual knowledge from vision. We learn them without forgetting the old knowledge.
> > >
> > > TLDR: Language models are NOT equipped with continual learning ability **without forgetting**; zero-shot problem is NOT equal to continual learning problem; existing zero-shot methods in the reference list do NOT address the problem of forgetting in continual learning
> > >
> > > Example: a pre-trained language model only knows to recognize <apples on the tables>. How does this language model learn to recognize <elephants In the jungle> without forgetting <apples on the tables>.
> > >
> > > We appreciate it if the reviewer can point out the existing work in continual learning to tackle this problem, i.e. learn new objects and new relationships without forgetting old objects and old relationships.
> > >
> > > ====
> > > Language models for zero-shot learning are generally NOT equipped to handle the challenges of continual learning. The primary reason is catastrophic forgetting. When a language model is updated or fine-tuned on new data, it often overwrites its existing knowledge, leading to a significant drop in performance on previously learned tasks. This is because traditional language models lack mechanisms to preserve old knowledge while integrating new information. Hence, current language models cannot help us eliminate forgetting problems in vision.
> > >
> > > In contrast, continual learning models are specifically designed to address this issue, often using techniques such as memory replay, regularization methods, or architectural modifications to retain past knowledge while learning from new data.
> > >
> > > So, while zero-shot learning allows for impressive generalization to unseen tasks, it does not address the ongoing challenge of learning from a continuous stream of data without forgetting. This is a fundamental limitation when applying zero-shot models to real-world scenarios requiring continual adaptation without forgetting.
> > >
> > > We will cite all the recommended papers in zero-shot literature and highlight the key differences between zero-shot and continual learning.

---

### Official Review · Reviewer_myHM · 2024-07-13

**Soundness:** 3
**Presentation:** 3
**Contribution:** 2
**Rating:** 6
**Confidence:** 4

**Summary:**

The paper introduces the problem set up of continual learning for image scene graph generation. To this end, they  reorganize existing SGG datasets to establish this new benchmarks on three learning scenarios. Next, they present a "Replays via Analysis by Synthesis" (RAS) for generate diverse scene structure, followed by synthesizing scene graphs for replays.

**Strengths:**

The scenarios and benchmarks experiments using both transformer-based and CNN-based backbones are exhaustive. The proposed RAS outperforms baselines for two scenarios.
RAS parses previous task scene-graphs into triplet labels for diverse in-context scene-graph reconstruction. RAS then synthesizes images with Stable Diffusion models for replays from the above re-compositional graphs.

**Weaknesses:**

For the continual learning setup, I'd imagine a video/dynamic scene graph generation will be more useful. However, currently it's focused on image scene graphs alone.

**Questions:**

What do the authors think about creating a similar benchmark for dynamic scene-graphs such as Action Genome or others?

**Limitations:**

Apart from simple geometric/locations, most important visual relations are associated with action, and those actions evolve over time. A good examples of such dynamic scene graphs would be action genome, HOMAGE similar, and more recently EASG. e.g. a single image may not be enough to understand if a person is getting down or getting up for riding a horse.

I'd like to understand the scenario for the continual learning setup can work for video/dynamic scene graphs.

[1] Ji. et al, Action Genome: Actions as Compositions of Spatio-temporal Scene Graphs, CVPR 2022
[2] Rai et al, "HomeAction Genome: Cooperative Compositional Action Understanding", CVPR 2021
[3] Rodin, Furnari, Min et al, "Action Scene Graphs for Long-Form Understanding of Egocentric Videos", CVPR 2024

---

> ### Author Rebuttal · Authors · 2024-08-06
>
> **[myHM.Weakness.1 - Focus on Image SGG explanation]** We agree with the reviewer that advancing continual learning in Scene Graph Generation (SGG) to include dynamic or video SGG is a natural and beneficial progression, as it more closely aligns with real-world settings.
>
> However, this research topic on continual learning in SGG is relatively unexplored, as no one has studied it before. Therefore, we need to begin with basic and straightforward settings, i.e. continual learning in scene graph generations on static images. Here, we aimed to establish a comprehensive benchmark of baselines and datasets to provide a foundation for the research community to start developing more advanced continual learning methods for SGG on static images. With all these foundations developed in this work,  we can then tackle more complex aspects, such as dynamic video SGG in the next phase.
>
> **[myHM.Questions.1 - Opinion on Dynamic SGG Benchmark]** We agree that creating a similar benchmark for dynamic scene-graphs is an excellent idea, and we definitely plan to extend our benchmark dataset and to incorporate dynamic scene-graph models. As mentioned in our previous response, we are the first team to address SGG in a continual learning setting, and our initial goal was to begin with basic and straightforward settings on static images. Only after this, we can move on to incorporate dynamic SGG models into our benchmark. Thank you for bringing Action Genome to our attention; it has inspired us for future developments in our benchmark. We will certainly cite this work in our final version.
>
> **[myHM.Limitations.1 - Continual learning for dynamic SGG]** We agree with the reviewer that advancing continual learning in Scene Graph Generation (SGG) to include dynamic or video SGG is a natural and beneficial progression, as it more closely aligns with real-world settings. As mentioned earlier, we are the first team to address SGG in a challenging continual learning setting, and our initial goal was to establish a comprehensive benchmark for image-level SGG models. Only from here, we can then move on to develop continual learning SGG in dynamic and video settings.
>
> We thank the reviewer for the clear directions in continual learning for dynamic scene graphs. In response, we will include a paragraph in the related works section to survey all the dynamic scene graph generation works, such as the three works here: Action Genome, HOMAGE, and EASG. We will cite these papers in the related works. However, do note that all these three works do not study continual learning problems. Neither do these works look into the importance of studying continual learning in SGG in a dynamic setting. Expanding our benchmark to incorporate continual learning in dynamic SGG models will be a key focus of our future work. Thank you for the valuable suggestions and for highlighting important works in this area.

---

### Official Review · Reviewer_Y6eG · 2024-07-22

**Soundness:** 2
**Presentation:** 2
**Contribution:** 3
**Rating:** 6
**Confidence:** 3

**Summary:**

This paper proposes a benchmark and a framework for incremental scene graph generation using continual learning. They curated the benchmark over an existing benchmark SGG dataset, VG. They proposed three learning scenarios such as relationship incremental,  scene incremental and generalization on relationship. Each of these learning regimes has different continual learning challenges such as learning new relationships, learning both new objects and relationship etc. The authors proposed a generative replay method which can deal with the catastrophic forgetting of continual learning. They reported detailed experimental results along with ablation studies.

**Strengths:**

1. Applying continual learning to scene graph generation could greatly benefit tasks like robotic navigation etc. where the agent need to adapt in new scenerios with new objects and relationships
3. The data and codes of the experiment will be publicly available

**Weaknesses:**

1. While the paper proposes a benchmark and framework for continual learning for SGG with detailed experiemtns and evaluation, the
presentation of the paper is very difficult to follow.

2. The problem formulation and formal mathematical definition of the overall methods are missing for the most of the parts. Section 3 has some definitions at the beginning and then explained learning scenarios and competitive baselines followed by the explanation of the evaluation metrics. Section 4 explains the continual learning framework 'Replays vai Analysis by Synthesis', but how this framework work in conjuction of SGG backbones (one-stage and two-stage existing SGG approach) needs to be clarified and formulated in formal equations.

3. For Gen Rbbox@K metrics, could you please share some details on how you calculated the metric without the labels and how you ran the inference for predicting the relationship?

**Questions:**

1. For Gen Rbbox@K metrics, could you please share some details on how you calculated the metric without the labels and how you ran the inference for predicting the relationship?

**Limitations:**

yes, the authors has addressed the limitations and potential negative societal impacts

---

> ### Author Rebuttal · Authors · 2024-08-06
>
> **[Y6eG.Weakness.1-Poor paper presentation]** We appreciate the reviewer’s feedback regarding the presentation of our paper. We would like to note that other reviewers did not mention any issues with the clarity or presentation of our work. However, we value your input and it would be great if you could specify which parts of the paper were difficult to follow. This would help us make the necessary revisions and improvements for the final version. Thank you for your constructive feedback.
>
> **[Y6eG.Weakness.2-Lack of Mathematical Formulation]** For each scenario in CSEGG, we have task-specific dataset $D_t \$, where  $t = 1, 2, 3, 4, 5$  for Learning Scenario 1 (S1), $t = 1, 2 $ for Learning Scenario 2 (S2), and $ t = 1, 2, 3, 4 $ for Learning Scenario 3 (S3).
>
> At task $ t $, we take the previously trained model $ M_{t-1} $ (with weights $ W_{t-1} $) and train it on the dataset for task $ t $, $ D_t $, to obtain weights $ W_t $ and model $ M_t $. If using an exemplar-based approach (Replays and RAS), the model is also trained on $ E_t $, where $ E_t $ is the exemplar for task $ t $. Note that $ E_1 $ doesn’t exist as no exemplar is needed for task 1 in all scenarios.
>
> For replay-based methods, we create exemplar $ E_t $ after training of task $ t-1 $ by storing data points from $ D_{t-1} $. Therefore, after training of task $ t $, we create $ E_{t+1} $ from $ D_t $. At task $ t+1 $, we obtain $ M_{t+1} $ by training $ M_t $ (with weights $ W_t $) on $ E_{t+1} $ and $ D_{t+1} $.
>
> For RAS, we create $ E_{t+1} $ as shown in Fig 3 in the manuscript. Following that, at task $ t+1 $, we obtain $ M_{t+1} $ by training $ M_t $ (with weights $ W_t $) on $ E_{t+1} $ and $ D_{t+1} $.
>
> If we are using EWC, we use $ W_t $ and $ M_t $ to calculate $ F_{t+1} $, where $ F $ is the Fisher information matrix using the equation, $ F_{t+1} = -E[\frac{\partial^2 }{\partial W_{t}^2}log(M_t(x)|W_t)] $, where $ M_t $ is the model of task t and $ W_t $ is the weights of task $ t $. Now during training of task $ t+1 $, we add $ L_{EWC} $ to the training loss. We calculate $ L_{EWC} $ using the equation, $ L_{EWC} = \sum_{}^{}F_{t+1}(W_{t+1} - W_{t})^2 $. While training for task $ t+1 $ we modify the loss $ L_{t+1}  = L_{train} +  L_{EWC} $.
>
> If we are using PackNet, after training of task $ t $ we take $ W_t $ and apply a pruning algorithm to obtain $ (W_t)’ $, where $ (W_t)’ $ are the pruned weights. At task $ t+1 $, we obtain $ M_{t+1} $ by training $ M_t $ with $ (W_t)’ $ and training on $ D_{t+1} $.
>
> We are also attaching a **Figure R1**, in the rebuttal pdf to show the entire schematic of CSEGG.
>
>
> **[Y6eG.Weakness.3-Gen Rbbox@K explanation]** Gen Rbbox@K is specifically defined for Learning Scenario 3 (S3) to evaluate the CSEGG model’s performance in locating bounding box locations. As described in Section 3.1, S3 consists of 4 tasks, with each task incrementally introducing 30 new objects. The relationship classes remain consistent across all tasks. In S3, there is a standalone test set where novel objects, absent from any training set of S3, appear with the same set of common relationship classes used in training.
>
> During evaluation, to calculate Gen Rbbox@K, for each predicted bounding box, we calculate the Intersection over Union (IoU) between the predicted bounding box and the ground truth boxes. If the IoU exceeds the predetermined threshold, it is considered a true positive (TP). We used IoU thresholds of 0.3, 0.5, and 0.7 for our experiments. If the IoU exceeds the threshold multiple times for the same predicted bounding box, we consider it a single positive prediction. This is because the metric aims to define the model’s performance in locating bounding boxes of objects that have not been learned during training. Counting multiple times for the same box would be misleading, as it would inflate the number of TPs and recall, while the actual number of unknown bounding boxes the model can generate might be low. The total possible positives (TP + FN) are determined by the total number of ground truth boxes in the image.
>
> This metric helps evaluate how well the CSEGG model locates unknown objects within an image. True positives (TP) represent successful identification of the location of an unknown object, while (TP + FN) represents all possible unknown objects the CSEGG model could locate. Thus, object classification labels are not required to calculate Gen Rbbox@K. Instead, we need the total number of ground truth bounding boxes in an image and the number of predicted boxes that meet the IoU threshold as explained in Section 3.3.
>
> **[Y6eG.Question.1- Gen Rbbox@K explanation]** Same as **[Y6eG.Weakness.3]**

---

> > ### Author Response · Authors · 2024-08-12
> > **Paper presentation and metric explaination**
> >
> > Dear reviewer,
> >
> > We wonder whether our comments have addressed your concern about the paper presentation and the clarification on the evaluation metric Gen Rbbox@K.
> >
> > We are open to further discussions if needed.

---

### Author Rebuttal · Authors · 2024-08-07

We appreciate all the reviewers' feedback. We encourage reviewers to refer to the PDF file containing additional figures. To differentiate these new figures in the rebuttal from those in the main text, we have prefixed them with "R" in the rebuttal. For example, Fig R1 corresponds to Fig 1 in the rebuttal PDF. We've included a point-by-point response for each of the three reviewers.

---

### Decision · Program_Chairs · 2024-09-25

**Decision:**

Accept (poster)

**Comment:**

The paper initially received mixed-to-negative reviews. While the reviewers appreciated the exhaustive experiments and the clarity in presentation, there were some concerns over (i) the problem statement, (ii) the limited explanation of the metrics, (iii) the lack of comparison with newer baselines, and (iv) the lack of fine-grained control signals. The authors' response addressed many of the concerns effectively. While concerns remain over the utility of the problem setting and somewhat limited baselines for two-stage networks, the paper offers an interesting perspective for SGG and will serve as a solid baseline for continual learning for SGG. The authors are strongly encouraged to incorporate the discussion from the rebuttal into the final version for completeness.